# Deliberation on Priors: Trustworthy Reasoning of Large Language Models on Knowledge Graphs

**Jie Ma**[*1†]**, Ning Qu**[*1,2]**, Zhitao Gao**[1,2]**, Rui Xing**[1]**, Jun Liu**[2,3]**, Hongbin Pei**[1]**,
**Jiang Xie**[4]**, Lingyun Song**[5]**, Pinghui Wang**[1]**, Jing Tao**[1]**, Zhou Su**[1]
[1]MOE KLINNS Lab, Xi'an Jiaotong University
[2]School of Computer Science and Technology, Xi'an Jiaotong University
[3]Shaanxi Province Key Laboratory of Big Data Knowledge Engineering
[4]School of Artificial Intelligence, Chongqing University of Post and Telecommunications
[5]School of Computer Science, Northwestern Polytechnical University
[*]Equal contribution
[†]Corresponding Author
jiema@xjtu.edu.cn

## Abstract

Knowledge graph-based retrieval-augmented generation seeks to mitigate hallucinations in Large Language Models (LLMs) caused by insufficient or outdated knowledge. However, existing methods often fail to fully exploit the prior knowledge embedded in knowledge graphs (KGs), particularly their structural information and explicit or implicit constraints. The former can enhance the faithfulness of LLMs' reasoning, while the latter can improve the reliability of response generation. Motivated by these, we propose a trustworthy reasoning framework, termed Deliberation over Priors (DP), which sufficiently utilizes the priors contained in KGs. Specifically, DP adopts a progressive knowledge distillation strategy that integrates structural priors into LLMs through a combination of supervised fine-tuning and Kahneman-Tversky optimization, thereby improving the faithfulness of relation path generation. Furthermore, our framework employs a reasoning-introspection strategy, which guides LLMs to perform refined reasoning verification based on extracted constraint priors, ensuring the reliability of response generation. Extensive experiments on three benchmark datasets demonstrate that DP achieves new state-of-the-art performance, especially a H@1 improvement of 13% on the ComplexWebQuestions dataset, and generates highly trustworthy responses. We also conduct various analyses to verify its flexibility and practicality. Code is available at https://github.com/mira-ai-lab/Deliberation-on-Priors.

## 1 Introduction

Large Language Models (LLMs) [1–4], distinguished by their massive parameter scale and training on vast amounts of diverse, unlabeled data, have demonstrated impressive capabilities across a wide range of natural language understanding and generation tasks. They have also achieved substantial success in practical applications, such as intelligent virtual assistants and customer service systems. However, recent research [5–7] has revealed that LLMs are prone to hallucinations, producing plausible-sounding but incorrect or outdated responses, especially in real-world scenarios. This issue, often stemming from insufficient or obsolete knowledge, can lead to serious consequences and undermine the reliability of LLMs in high-stakes domains such as legal decision-making and medical diagnosis [8, 9].

39th Conference on Neural Information Processing Systems (NeurIPS 2025).

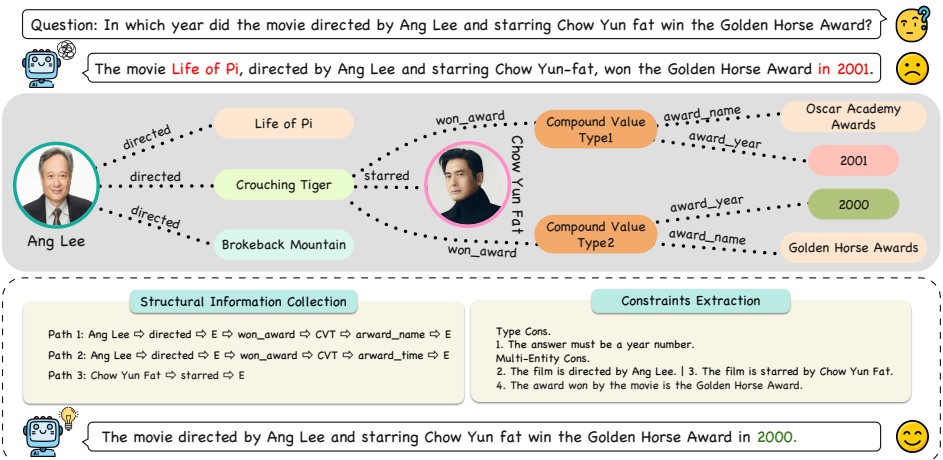

Figure 1: An illustration of LLM reasoning over knowledge graphs based on exploiting priors sufficiently. We collect weak supervision signals of the mapping from questions to relation paths by identifying the shortest traversal sequence from topic entities to answers. We predefine 5 constraints, such as type, multi-entity, and ordinal constraints, following [17] which employs them to develop the ComplexQuestions dataset but does not leverage the prior in reasoning.

To equip LLMs with up-to-date, domain-specific knowledge, an emerging line of research [10–12] has focused on knowledge graph-based retrieval-augmented generation, which aims to enhance response generation through the dynamic retrieval of relevant external knowledge. Current methods [7, 8, 13, 14] employ an end-to-end or step-by-step manner to retrieve and reason on Knowledge Graphs (KGs). The former retrieves the top $K$ triples based on the semantic similarity between questions and knowledge facts, while the latter transforms complex questions into multiple subquestions by step-by-step retrieval. After obtaining sufficient knowledge, they utilize LLMs to generate responses directly.

However, existing approaches [7, 8, 13–16] fail to exploit the prior knowledge embedded in KGs fully, particularly (1) *the structural information* and (2) *the explicit and implicit constraints*. In particular, relation paths that connect topic entities in questions to their corresponding answers, such as the path "Ang Lee → directed → ... E" in Figure 1, can be leveraged not only to enhance the structural pattern awareness within KGs but also to support response generation. This, in turn, can significantly improve the faithfulness of LLM reasoning over KGs. Additionally, constraints, such as the multi-entity and type constraints, can serve a dual purpose: they can be used to filter candidate relation paths (select path 2 in Figure 1 considering the mentioned two types of constraints) and also to guide LLMs in backtracking during inference, thereby improving the reliability and robustness of reasoning processes. Motivated by these, we propose a trustworthy reasoning framework over KGs named DP (Deliberation on Priors). The framework comprises four key modules: *Distillation, Planning, Instantiation, and Introspection*, which guide LLMs to generate faithful and reliable responses through a two-stage process: offline and online.

In the offline stage, DP first collects weak supervision signals in the form of mappings from questions to corresponding sets of relation paths. The paths are constructed by identifying the shortest traversal sequences from topic entities to answer entities in KGs. This prior structural knowledge is then *distilled* into the LLM through a combination of fine-tuning and Kahneman-Tversky optimization [18]. The former is supervised using the weak supervision signal, while the latter is optimized by maximizing the expected utility of generation under the human utility model that Kahneman and Tversky proposed to describe how humans make decisions about uncertain monetary outcomes. This process enhances the ability of LLMs to plan and reason over the KG structure effectively.

During the online stage, the trained LLM is used to *Plan* for reasoning and generate faithful candidate relation paths. Subsequently, DP utilizes LLMs to perform relation path selections by evaluating the semantic relevance between the candidate path and the input question. Next, to obtain a complete reasoning path, the framework retrieves topic entities and relations from KGs and instantiates the selected relation path in the *Instantiation* module, such as the transformation from "{directed, won_award}"

Table 1: Constraint definitions and samples.

| Category | Definition & Sample |
|---|---|
| Type | The question specifies a required type or category for the answer. 
 *e.g.* What `city` did Esther live in? |
| Multi-Entity | The question demands the answer to satisfy the condition for multiple entities. 
 *e.g.* Which team owned by `Malcolm Glazer` has `Tim Howard` playing for it? |
| Explicit Time | The question clearly defines a specific time period or date to be referenced. 
 *e.g.* Who was the governor of Arizona `in 2009` that held his governmental position `before 1998` ? |
| Implicit Time | The question implies a temporal frame that the answer should consider. 
 *e.g.* Who was the Secretary of State `when Andrew Jackson was the president` ? |
| Ordinal | The question contains the sorting rule and requires the answer in a specific order. 
 *e.g.* Lou Seal is the mascot for the team that `last` won the World Series when? |

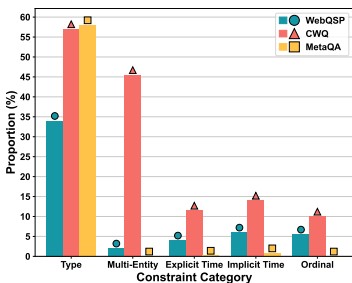

Figure 2: Constraint distribution on three datasets.

to "Ang Lee $\xrightarrow{\texttt{directed}}$ Crouching Tiger $\xrightarrow{\texttt{won\_award}}$ Compound Value Type 2". After obtaining the reasoning path, current frameworks [7, 8, 13–16] utilize LLMs to generate responses based on them without deliberation. In contrast, to enhance reasoning reliability, DP employs the *Introspection* module to verify whether the selected reasoning path satisfies the constraints extracted from questions, where the constraints, such as multi-entity and ordinal constraints, are predefined following [17]. Feedback is then provided to the LLM to guide subsequent decisions. If the reasoning path passes the verification, the LLM is instructed to generate a response. Otherwise, the information about which constraints are violated is fed back to the LLM to trigger relation path backtracking.

To evaluate the effectiveness of DP, we conduct extensive experiments on three benchmark Knowledge Graph Question Answering (KGQA) datasets. The experimental results on WebQuestionSP [19], ComplexWebQuestions [20], and MetaQA [21] show that, compared with baselines, our method achieves state-of-the-art performance while minimizing interaction frequency with LLMs and generates more faithful and reliable responses.

The main contributions of this work are summarized as follows:

- We introduce a trustworthy reasoning framework DP that empowers large language models to generate faithful and reliable responses through deliberate reasoning over the priors embedded in knowledge graphs.

- We propose a progressive knowledge distillation strategy, enabling LLMs to generate faithful relation paths by exploiting prior structural information.

- We propose a reasoning-introspection strategy that enhances the reliability of large language model generation by incorporating predefined constraint priors as guidance.

- We conduct extensive experiments on three public datasets to verify the effectiveness and superiority of DP. Furthermore, we demonstrate the flexibility of DP in integrating with various large language models, as well as its practicality in scenarios requiring fewer interactions with them.

## 2 Preliminary

**KGQA Task Formulation.** Given a question $q$, a knowledge graph $\mathcal{G}$, and a topic entity $e_s$ mentioned in $q$, KGQA requires the intelligent system, such as LLMs, to generate responses $a$ based on a set of knowledge triples (facts) retrieved from $\mathcal{G}$. It should be noted that question $q$ may contain multiple topic entities.

**Relation Path Definition.** A relation path is formally defined as an ordered sequence of relations $P = \{r_1, r_2, \ldots, r_l\}$, where each $r_i \in \mathcal{R}$ denotes the $i$-th relation in the path and $l$ represents the path length. Here, $\mathcal{R}$ denotes the set of all possible relations in $\mathcal{G}$.

**Path Instantiation.** Given a relation path $P$, an instantiation path is called a reasoning path, denoted by $\mathbb{P} = e_0 \xrightarrow{r_1} e_1 \xrightarrow{r_2} e_2 \xrightarrow{r_3} \cdots \xrightarrow{r_l} e_l$, where each $e_i \in \mathcal{E}$ denotes the $i$-th entity in the path and $r_i$ corresponds to the $i$-th relation in the relation path $P$. Here, $\mathcal{E}$ represents the set of all entities in the KG. It should be noted that a relation path $P$ may have multiple instantiations in $\mathcal{G}$.

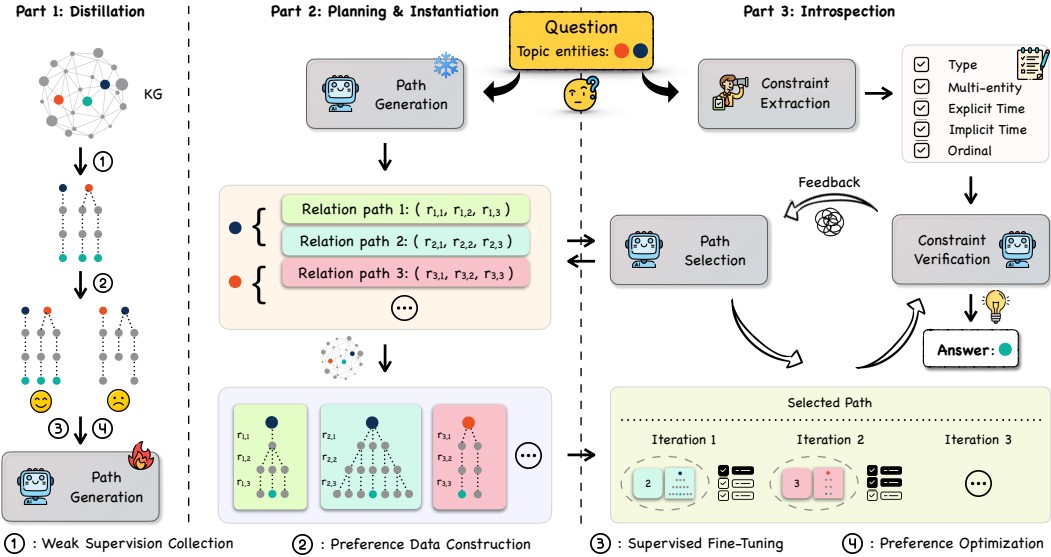

Figure 3: Trustworthy reasoning framework DP of LLMs over KGs. In *Part 1*, DP employs a progressive knowledge distillation strategy to enhance the structural pattern awareness of KGs for LLMs. In *Part 2*, the reasoning path is produced by relation path generation and instantiations. In *Part 3*, DP utilizes a reasoning-introspection strategy to verify whether the reasoning satisfies the extracted constraint.

**Constraint Predefining.** The constraints listed in Table 1 were initially introduced by [17] to facilitate the development of complex questions. In this paper, we leverage these constraint priors for reasoning verification. Specifically, we utilize multiple constraints like "type" and "multi-entity" extracted from questions in Figure 1 to assess whether a selected reasoning path satisfies the required conditions. To this end, we define a constraint base $C$ and conduct a statistical analysis across three datasets, sampling 200 examples from each. Figure 2 demonstrates that most datasets encompass a range of constraint categories, with the "type" constraint being the most frequently occurring in each.

## 3 Methodology

Our proposed framework, DP, as illustrated in Figure 3, is composed of four key components. *Distillation*: This module employs a progressive knowledge distillation strategy to guide LLMs in capturing the structural patterns of KGs from a set of demonstrations. *Planning and Instantiation*: In this stage, DP first generates a diverse set of candidate relation paths, which are subsequently grounded into KG triples to obtain instantiated paths. *Introspection*: This component performs a deliberative selection and verification of reasoning paths by evaluating whether they meet the constraints derived from the input question. Once a satisfactory instantiated path is identified, the LLM is employed to generate responses accordingly.

### 3.1 Distillation

DP employs a progressive knowledge-distillation strategy to guide LLMs in exploiting the structural information and enabling faithful reasoning over KGs. This is achieved by a combination of Supervised Fine-Tuning (SFT) and Kahneman-Tversky Optimization (KTO) [18]. The former fine-tunes the LLM using collected weak supervision signals, while the latter further refines it based on automatically derived preference data.

Specifically, DP leverages weak supervision signals in the form of question-to-path mappings, which are derived from the annotations within training splits. Given a question $q$ and a topic entity $e_s$ identified in $q$, we define a function $\mathcal{M}$ that maps $q \in \mathcal{Q}$ to a set of plausible relation paths $\mathcal{P}$, where each $P \in \mathcal{P}$ is a sequence of relations such that traversing from $e_s$ via $(r_1, \ldots, r_l)$ leads to the answer

entity $e_t$ in the KG $\mathcal{G} = (\mathcal{E}, \mathcal{R})$. To extract these paths, our framework first extracts a $k$-hop subgraph $\mathcal{G}_k(e_s)$ centered on $e_s$, where $k$ is the maximum reasoning depth allowed by the dataset. Then, the Dijkstra algorithm is employed to identify all shortest relation paths from $e_s$ to the ground-truth answer entity $e_t$ within $\mathcal{G}_k(e_s)$. These shortest paths constitute the weak supervision set $\mathcal{P}_w(q)$ for question $q$, serving as structurally mapping relation exemplars. While not exhaustive, $\mathcal{P}_w(q)$ captures plausible multi-hop reasoning trajectories within the KG structure and provides guidance for the LLM to generalize to similar questions. The weak supervision of question-to-path mappings $\mathcal{P}_w(q)$ can be defined as follows:

$$\mathcal{P}_w(q) = \mathcal{M}(q, e_s, \mathcal{G}, k) = \text{ShortestPath}_{\text{Dijkstra}}(\mathcal{G}_k(e_s), e_s, e_t). \tag{1}$$

We then apply SFT to train the LLM to generate relation paths conditioned on the input question $q$ and its corresponding topic entity $e_s$. This training stage encourages LLMs to align the semantic content of the question with structural relation traversals over $\mathcal{G}$. Let $P^* = (r_1^*, r_2^*, \ldots, r_T^*)$ denote a gold relation path extracted using Equation (1) for question $q$. The objective of SFT is to maximize the conditional log-likelihood $\mathcal{L}_{\text{SFT}}$ of the target path sequence $P^*$ given the input $(q, e_s)$:

$$\mathcal{L}_{\text{SFT}}(\theta_s) = \sum_{t=1}^{T} \log P_{\theta_s}\left(r_t^* \mid r_{<t}^*, q, e_s\right), \tag{2}$$

where $\theta_s$ represents the parameters of the LLM and $r_{<t}^*$ denotes the prefix subsequence $(r_1^*, \ldots, r_{t-1}^*)$. This formulation enables the LLM to learn to autoregressively generate faithful relation paths that connect $e_s$ to plausible answer entities within the KG.

To further enhance the reliability of relation path generation, we incorporate a preference optimization stage to explicitly encourage LLMs to prefer semantically coherent and structurally faithful relation paths. Specifically, we construct relation path data $D$ consisting of positive paths and negative paths given question $q$. The negative or undesirable path is synthetically generated from the original weak supervision data $\mathcal{P}_w(q)$ through targeted perturbations for the positive path:

- **Path Truncation.** Removing the final hop $r_T^*$ from a gold relation path $P^*$ to yield an incomplete relation chain.
- **Entity-Path Swapping.** Swapping relation paths between different topic entities associated with the same question, resulting in semantically inconsistent paths.
- **Relation Deletion.** Randomly deleting the relation path of a certain topic entity, resulting in incomplete paths.

These synthetic negative paths are constructed to superficially valid relation paths while being semantically invalid due to violations of critical structural or semantic constraints. Notably, such perturbations introduce severe class imbalance in path data, where positive paths constitute only $\frac{1}{4}$ of the dataset, and negative paths account for $\frac{3}{4}$. This imbalance poses significant challenges to conventional preference optimization like direct preference optimization. Therefore, we train the LLM using KTO [18], a more robust approach that accommodates imbalanced supervision. KTO maximizes the expected utility of generation under the human utility model that Kahneman and Tversky proposed to describe how humans make decisions about uncertain monetary outcomes. This utility maximization objective is equivalent to minimizing the KTO loss $\mathcal{L}_{\text{KTO}}$:

$$\mathcal{L}_{\text{KTO}}(\pi_\theta, \pi_{\text{ref}}) = \mathbb{E}_{(x,y) \sim D}\left[\lambda_y - v(x, y)\right]. \tag{3}$$

Here, we define the key components of the KTO loss formulation. Let $x = (q, e_s)$ denote the input, and let $y$ represent its corresponding relation path label, where $y \sim Y_p$ and $y \sim Y_n$ indicate a positive and negative label, respectively. $\pi_\theta(\cdot)$ is the current LLM's output distribution and $\pi_{\text{ref}}(\cdot)$ is the reference model supervised by SFT. The parameter $\lambda_y$ takes the value $\lambda_p$ (for positive $y$) or $\lambda_n$ (for negative $y$), controlling class-specific weighting. The value function $v(\cdot)$ models human utility perception, defined as:

$$v(x, y) = \begin{cases} \lambda_p \sigma\left(\beta(r_\theta(x, y) - z_0)\right) & \text{if } y \sim Y_p \mid x, \\ \lambda_n \sigma\left(\beta(z_0 - r_\theta(x, y))\right) & \text{if } y \sim Y_n \mid x. \end{cases} \tag{4}$$

where $\sigma(\cdot)$ denotes the logistic function, $\beta$ modulates risk aversion, $r_\theta(x, y) = \log \frac{\pi_\theta(y|x)}{\pi_{\text{ref}}(y|x)}$ represents the implied reward and the reference point $z_0$ is obtained via the KL divergence $z_0 = \text{KL}\left(\pi_\theta(y' \mid x) \,\|\, \pi_{\text{ref}}(y' \mid x)\right)$. In brief, the progressive knowledge distillation strategy equips DP with reliable structural priors of KGs, enabling faithful online reasoning in the subsequent stages.

## 3.2 Planning and Instantiation

In this stage, DP utilizes the path generator, which has been trained through the aforementioned progressive knowledge-distillation strategy, to generate multiple candidate relation paths. One of these paths is then selected and instantiated by retrieving entities from the KG, beginning with the given topic entity $e_s$. Importantly, the selection of the relation path at this stage is guided solely by its semantic alignment with the input question, without imposing any constraints.

More formally, given a question $q$ and its associated topic entity $e_s$, DP invokes the path generator to produce a set of candidate relation paths $\{P_1, P_2, \dots\}$, where each $P_i$ denotes a traversal sequence of relations representing a multi-hop reasoning trajectory over the KG. In scenarios where a question contains multiple topic entities, the path generator independently produces candidate paths for each entity. These paths are subsequently merged into a unified pool, thereby enriching semantic diversity and increasing the probability that at least one reasoning trajectory satisfies the implicit or explicit constraints embedded in the question. For a given relation path $P$, the corresponding instantiated reasoning path $\mathbb{P}$ is obtained by traversing the KG starting from the topic entity $e_s$. The instantiation process is described in detail in Section 2.

## 3.3 Introspection

To ensure the reliability of LLM reasoning, our framework utilizes a reasoning-introspection strategy to verify whether the reasoning path satisfies the extracted constraint from questions. As introduced in Section 2, we predefine 5 constraint types, which are prior knowledge embedded in KGs and can be employed to guide LLMs. Specifically, given question $q$, DP first prompts LLMs to extract the contained constraint $\mathcal{C}(q)$ from the predefined constraint base $C$: $\mathcal{C}(q) = \mathcal{F}_{\text{cons}}(q, C, I_{\text{cons}})$, where $I_{\text{cons}}$ represents the prompt and in-context exemplars. Subsequently, the LLM is prompted to determine whether the instantiated reasoning path $\mathbb{P}$ satisfies $\mathcal{C}(q)$ given $q$ and $e_s$. The verification outcome is formalized as:

$$\mathcal{J}(q, e_s, \mathbb{P}) = \begin{cases} 1, & \text{if } \mathbb{P} \models \mathcal{C}(q); \\ 0, & \text{otherwise.} \end{cases} \tag{5}$$

If the constraint is satisfied, DP instructs the LLM to generate a reason and produce a final response grounded on the validated reasoning path. Conversely, if the constraint is violated, the LLM is prompted to provide explicit feedback identifying the unsatisfied condition. This feedback subsequently triggers a backtracking mechanism, wherein the framework iteratively re-executes the process of relation path selection, instantiation, and introspection. This mechanism can reduce the negative impact of false-positive reasoning paths, promoting the reliability of response generation. The iterative loop continues until either a constraint-satisfying reasoning path is found or the candidate relation path set is reduced to a singleton.

# 4 Experiment

We conduct various experiments on three benchmarks to verify the following aspects: (1) the superiority of DP; (2) the flexibility in integrating different LLMs; (3) the effectiveness of individual components; (4) the necessity of deliberation on prior knowledge; and (5) the practicality of DP.

## 4.1 Experimental Setting

**Dataset and Evaluation Metric.** We evaluate our approach on three public multi-hop KGQA datasets: WebQuestionSP (WebQSP) [19] and ComplexWebQuesions (CWQ) [20], and MetaQA [21]. WebQSP is developed by gathering semantic parses in SPARQL from the WebQuestions dataset. CWQ features a large collection of compositional questions, requiring reasoning over up to 4-hop relation paths based on multiple web snippets. MetaQA is built upon a movie ontology derived from the WikiMovies dataset and provides question-answer pairs spanning 1-hop, 2-hop, and 3-hop queries. The KGs used in WebQSP and CWQ are a subset of Freebase. To account for computational constraints, we uniformly sample 500 questions from the test set of WebQSP and CWQ, following the setup in [22, 15]. A total of 600 instances are uniformly sampled from the MetaQA dataset, with exactly 200 samples selected for each of the 1-hop, 2-hop, and 3-hop types. Evaluation is conducted using three standard metrics: Hit, Hits@1, and F1 score, consistent with prior studies [8, 23, 24]. The

Table 2: Comparison of KGQA performance (%) across three datasets with previous state-of-the-art methods. DP results are averaged over three independent evaluations. We highlight the best performance in bold and the second-best in underline. Baseline methods are categorized into three groups: Supervised Learning (SL), In-Context Learning (ICL), and Hybrid Learning (HL). LM denotes the language model used. ⋆ denotes the performance on 3-hop questions.

| Type | Method | Year | LM | WebQSP | | | CWQ | | | MetaQA | | |
|------|--------|------|-----|--------|------|------|------|------|------|--------|------|------|
| | | | | H | H@1 | F1 | H | H@1 | F1 | H | H@1 | F1 |
| Vanilla | Zero-shot | 2022 | ChatGPT | 59.3 | 59.3 | 43.5 | 34.7 | 34.7 | 30.2 | - | - | - |
| | | 2023 | GPT-4 | 67.4 | 67.4 | 49.5 | 50.8 | 50.8 | 46.1 | - | - | - |
| | | 2024 | LLaMA3.1-8B | 55.5 | 55.5 | 34.8 | 28.1 | 28.1 | 22.4 | - | - | - |
| SL | EmbedKGQA | 2020 | RoBERTa | - | 66.6 | - | - | - | - | - | 97.0 | - |
| | TransferNet | 2021 | BERT | - | 71.4 | - | - | 48.6 | - | - | 99.2 | - |
| | UniKGQA | 2023 | RoBERTa | - | 77.2 | 72.2 | - | 51.2 | 49.0 | - | **99.4** | - |
| | RoG | 2024 | LLaMA2-Chat-7B | 85.7 | 80.8 | 70.8 | 62.6 | 57.8 | 56.2 | - | 89.0* | 50.7* |
| | AMAR | 2025 | LLaMA2-7B/13B | 84.2 | - | 81.2 | 83.1 | - | **78.5** | - | - | - |
| | GNN-RAG | 2025 | LLaMA2-Chat-7B | 90.7 | 82.8 | 73.5 | 68.7 | 62.8 | 60.4 | - | 98.6* | - |
| ICL | ToG | 2024 | ChatGPT | 76.2 | - | - | 58.9 | - | - | - | - | - |
| | | | GPT-4 | 82.6 | - | 36.4 | 69.5 | - | 31.8 | - | - | - |
| | PoG | 2024 | ChatGPT | 82.0 | - | - | 63.2 | - | - | - | - | - |
| | | | GPT-4 | 87.3 | - | - | 75.0 | - | - | - | - | - |
| | Readi | 2024 | ChatGPT | 74.3 | - | - | 55.6 | - | - | - | - | - |
| | | | GPT-4 | 78.7 | - | - | 67.0 | - | - | - | - | - |
| | DoG | 2025 | ChatGPT | 88.6 | 62.6 | 54.2 | 58.2 | 49.4 | 56.6 | 95.4 | 85.1 | 87.2 |
| | | | GPT-4 | **91.0** | 65.4 | 55.6 | 56.0 | 41.0 | 46.4 | 98.3 | 90.1 | 93.1 |
| HL | Interactive-KBQA | 2024 | GPT-4 | - | - | 71.2 | - | - | 49.1 | - | - | **96.3** |
| | LightPROF | 2025 | LLaMA3-8B | - | 83.8 | - | - | 59.3 | - | - | - | - |
| | DP (Ours) | 2025 | LLaMA3.1-8B | 87.9±0.2 | 82.8±0.4 | 75.7±0.3 | 70.8±0.3 | 61.1±0.4 | 58.5±0.9 | 90.2±0.1 | 87.4±0.2 | 84.1±0.5 |
| | | | ChatGPT | 89.7±0.6 | $\underline{86.9}$±0.3 | 79.2±0.4 | 80.0±0.6 | 72.6±0.1 | 69.2±0.4 | 96.7±0.1 | 95.4±0.3 | 90.8±0.4 |
| | | | GPT-4 | 90.4±0.4 | 86.7±0.5 | **81.7**±0.6 | 85.6±0.3 | $\underline{74.6}$±0.7 | 71.1±0.8 | 96.7±0.2 | 95.4±0.4 | 94.8±0.2 |
| | | | GPT-4o | $\underline{90.7}$±0.6 | **87.5**±0.8 | 81.4±0.5 | $\underline{85.2}$±0.4 | $\underline{74.6}$±0.5 | 70.5±0.5 | 96.5±0.6 | 95.2±0.4 | 94.4±0.4 |
| | | | GPT-4.1 | 90.6±0.5 | 86.7±0.4 | $\underline{80.1}$±0.8 | **87.2**±0.2 | **75.8**±0.7 | $\underline{69.4}$±0.9 | $\underline{96.8}$±0.2 | 95.5±0.3 | $\underline{94.9}$±0.1 |

Hit (H) metric assesses whether any of the ground-truth answers are present in the generated response. Hits@1 (H@1) measures the proportion of questions for which the top-ranked predicted answer exactly matches a correct answer. The F1 score accounts for scenarios with multiple correct answers by computing the harmonic mean of precision and recall, thereby providing a more comprehensive evaluation of answer quality.

**Implementation Detail.** During the *Distillation* stage, the path generator (LLaMA3.1-8B-Instruct) is fine-tuned by the weak supervision signal in the form of question-to-path mapping for 2 epochs and further optimized for 1 epoch through preference optimization with KTO [18]. We apply low-rank adaptation [25] to adapt large-scale parameters during both SFT and KTO training efficiently. In the *Planning* stage, we employ the trained path generator in a zero-shot manner to produce a set of relation paths for each topic entity in the question. In the *Introspection* stage, DP guides LLMs to perform path selection, constraint extraction, and verification under a few-shot setting. Specifically, we provide one exemplar for each possible scenario within these three procedures, which consist of 3, 5, and 2 scenarios, respectively. The detailed settings are provided in Appendix B.1 and B.2.

**Baseline Selection.** Inspired by the setup in [15, 8], we compare DP against previous state-of-the-art approaches, including Supervised Learning (SL), In-Context Learning (ICL), and Hybrid Learning (HL). SL-based methods train models using the answer labels provided in KGQA datasets to directly predict answers, including EmbedKGQA [26], TransferNet [27], UniKGQA [28], RoG [8], AMAR [10], and GNN-RAG [24]. ICL-based methods employ chain-of-thought reasoning to generate responses based on few-shot exemplars, including PoG [29], ToG [22], Readi [30], and DoG [15]. HL-based methods combine SL and ICL to produce responses, including Interactive-KBQA [31] and LightPROF [32].

## 4.2 Reasoning on Different KGs

**Main Result.** We compare DP with previous state-of-the-art methods, categorized into three types: SL, ICL, and HL. Based on the results presented in Table 2, we draw the following key insights. First, incorporating deliberation over prior knowledge significantly enhances the reliability of response generation. The results for DP are averaged over three independent runs. Our framework consistently achieves new state-of-the-art performance across most datasets, with low variance. In contrast, the vanilla LLMs (without DP) exhibit unstable performance and often fail to achieve competitive H@1 or F1 scores, particularly on CWQ and WebQSP, highlighting their limitations in complex reasoning tasks. Notably, DP surpasses the HL-based method LightPROF by 16.5% in H@1 on the CWQ

dataset. Second, existing methods struggle in scenarios where the correct answer must appear in the top-ranked position or where the output is expected to contain accurate answers with high precision. For instance, ToG exhibits a 46.2% gap between its H and F1 scores on the WebQSP dataset. In contrast, DP narrows this gap to approximately 10%, further underscoring its robustness and reliability. Third, we observe that some prior works have misinterpreted evaluation metrics such as H and H@1. According to the code released by [10, 22], certain results reported as H@1 are actually H, potentially leading to unfair comparisons and misleading evaluations. In this work, we rigorously report DP's performance using H, H@1, and F1 metrics, respectively. Overall, the DP framework demonstrates robust and reliable performance across diverse datasets, validating the effectiveness of incorporating prior knowledge deliberation.

**Flexibility Verification.** Table 2 also reports the integration experiments, which are designed to evaluate whether DP can enhance the reasoning reliability of various LLMs, including LLaMA3.1-8B, GPT-3.5, GPT-4.0, GPT-4o, and GPT-4.1. The results show that DP consistently improves the reasoning performance of these LLMs across three benchmark datasets. Notably, our framework enables GPT-4o and GPT-4.1 to achieve the best performance on the WebQSP and CWQ datasets, respectively. Overall, these findings highlight the flexibility and effectiveness of DP in enhancing diverse LLMs, with lower variance observed across multiple runs, indicating that the improvements are stable and reliable.

## 4.3 Ablation Study

We perform comprehensive ablation studies on two benchmark datasets (WebQSP and CWQ) to systematically evaluate the contribution of each DP component. As shown in Table 3, our analysis reveals five critical observations. First, LLMs such as GPT-4.1 demonstrate significant performance degradation when deprived of knowledge augmentation. Specifically, the F1 score drops by 25.5% on WebQSP and 20.5% on CWQ, highlighting the necessity of external knowledge integration for complex question answering tasks. Second, our progressive knowledge distillation strategy effectively promotes the prior awareness of LLMs for the structural patterns of KGs. When removing KTO, we observe a 2.0% decrease in H@1 score on WebQSP and a 1.2% decrease on CWQ, confirming its role in improving pattern recognition capa-

Table 3: Ablation experiments. PT, EPS, and RD denote path truncation, entity-path swapping, and relation deletion, respectively, which are perturbations introduced in Section 3.1. "w/o CPD" indicates the constraint is automatically induced by LLMs rather than being manually predefined.

| Setting | WebQSP | | | CWQ | | |
|---|---|---|---|---|---|---|
| | H | H@1 | F1 | H | H@1 | F1 |
| DP (GPT 4.1) | **90.6** | **86.7** | **80.1** | **87.2** | **75.8** | **69.4** |
| GPT 4.1 | 74.0 | 71.0 | 54.6 | 56.0 | 53.0 | 48.9 |
| w/o KTO | 88.3 | 84.7 | 77.3 | 86.0 | 74.6 | 67.3 |
| w/o PT | 88.4 | 84.8 | 77.6 | 86.4 | 75.0 | 68.4 |
| w/o EPS | 90.0 | 85.8 | 79.4 | 86.6 | 73.8 | 68.1 |
| w/o RD | 88.3 | 84.8 | 77.8 | 87.2 | 74.1 | 67.7 |
| w/o Introspection | 86.6 | 82.0 | 75.7 | 85.3 | 70.8 | 65.2 |
| w/o CPD | 87.8 | 83.4 | 76.4 | 86.2 | 74.4 | 68.5 |
| w/o feedback | 88.0 | 83.0 | 76.5 | 85.4 | 73.2 | 67.1 |

bilities. Third, among the three perturbation methods examined, relation deletion demonstrates the most substantial impact on path generation. Fourth, the reasoning-introspection strategy proves to be the most critical component for ensuring response reliability. Compared to other ablated conditions, removing introspection leads to the most pronounced performance deterioration across both datasets. Finally, both constraint predefinition and verification feedback contribute significantly to the overall reasoning process. Notably, substituting manual constraint definitions with LLM-generated summaries results in a 2.8% drop in H score on WebQSP and a 1.0% decrease on CWQ, underscoring the value of human-defined constraints in guiding accurate reasoning. These findings collectively demonstrate the complementary roles of different DP components in enhancing the faithfulness and reliability of LLM reasoning.

## 4.4 Impact of Priors on LLM Reasoning

To investigate the impact of prior knowledge on LLM reasoning, we conduct extensive experiments on two benchmark datasets: WebQSP and CWQ. First, Table 4 presents the results of path generation and constraint extraction on 500 and 100 uniformly sampled examples from each dataset, respectively. To improve the faithfulness of LLM reasoning, we collect all relation paths that satisfy the condition defined in Equation (1) for each question. This leads to a one-to-many question-to-relation mapping, in contrast to the one-to-one mapping adopted by RoG [8]. Our approach yields significant improvements in the F1 score of path generation, with relative gains of 29.3% on WebQSP and 43.0%

Table 4: Results of Path Generation (PG) and Constraint Extraction (CE).

| Setting | WebQSP | | CWQ | |
|---|---|---|---|---|
| | H | F1 | H | F1 |
| PG (1: 1) | 83.0 | 59.3 | 81.2 | 49.8 |
| PG (1: n) | 93.0 | 76.7 | 94.0 | 71.1 |
| CE | 99.0 | 90.2 | 99.0 | 92.9 |

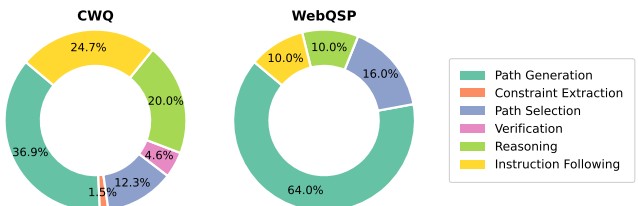

Figure 4: Error distribution on two datasets.

on CWQ, demonstrating the benefits of leveraging priors. Second, we analyze the failure cases of DP with GPT-4.1 in the main experiment. As shown in Figure 4, the majority of errors stem from path generation, path selection, reasoning, and instruction-following. Among these, path generation and selection are closely tied to the utilization of priors, underscoring the importance of effectively incorporating prior knowledge. Reasoning errors, despite the correct path selection, indicate that LLMs may rely excessively on their internal knowledge during response generation. Moreover, we observe that LLMs occasionally fail to adhere to the required answer format—for instance, producing "2008" instead of "2008 World Cup." We found that incorporating instruction-following exemplars mitigates this issue and yields performance improvements, such as a 1.1% Hit@1 gain on CWQ.

## 4.5 Backtracking Analysis

Backtracking refers to the iterative re-execution of relation path selection, instantiation, and introspection when a constraint violation is detected. To explore the practical impact of backtracking, we measure how often it is triggered during test time. Table 5 reports the average number of backtracking steps per question across CWQ and WebQSP. We observe that GPT-4.1 triggers backtracking more frequently than GPT-3.5 (e.g., 0.21 vs. 0.07 on WebQSP), likely due to its stronger instruction-following ability, which enforces stricter constraint checking. This aligns with the performance differences presented in Table 2, suggesting that more reliable backtracking contributes to better overall reasoning quality.

Table 5: Average number of backtracking steps per question across datasets and models.

| Method | Dataset | Model | Backtrack |
|---|---|---|---|
| DP | CWQ | GPT-3.5 | 0.10 |
| | | GPT-4.1 | 0.42 |
| | WebQSP | GPT-3.5 | 0.07 |
| | | GPT-4.1 | 0.21 |

## 4.6 Analysis for LLM Interaction

We conduct experiments on CWQ and WebQSP to evaluate the practicality and efficiency of DP. Table 6 presents a comparative analysis of the average number of LLM calls and token consumption required by different methods to answer a question across the two datasets. The results for DoG and DP are obtained using GPT-3.5, while additional results of DP with other LLMs are provided in Appendix B.5. Across both datasets, DP consistently outperforms strong baselines across all evaluation metrics. Specifically, our framework requires only 2.9 and 2.5 LLM calls on CWQ and WebQSP, respectively—highlighting its efficient utilization of prior knowledge embedded in KGs. In terms of token consumption, DP achieves the lowest

Table 6: Practicality and efficiency Comparison. Call denotes the number of LLM interactions. Input, Output, and Total represent the number of corresponding tokens. The results of DP are averaged over three independent runs.

| Dataset | Method | Call | Input | Output | Total |
|---|---|---|---|---|---|
| CWQ | ToG | 22.6 | 8,182.9 | 1,486.4 | 9,669.4 |
| | PoG | 13.3 | 7,803.0 | 353.2 | 8,156.2 |
| | DoG | 12.7 | 8,298.0 | 384.6 | 8,682.6 |
| | DP | **2.9** | **2,928.6** | **186.4** | **3,115.0** |
| WebQSP | ToG | 15.9 | 6,031.2 | 987.7 | 7,018.9 |
| | PoG | 9.0 | 5,234.8 | 282.9 | 5,517.7 |
| | DoG | 10.4 | 7,332.1 | 277.2 | 7,604.3 |
| | DP | **2.5** | **2,552.8** | **146.7** | **2,699.5** |

output token count while also requiring the least input token consumption. This reflects DP's effectiveness in reducing overall token usage during the reasoning process, relying on fewer instructions and exemplars. These findings underscore the superiority of DP compared to existing baselines.

# 5    Conclusion and Limitation

This paper presents a trustworthy reasoning framework, DP, which empowers Large Language Models (LLMs) to generate faithful and reliable responses by deliberately reasoning over the priors embedded in knowledge graphs. In the offline stage, DP enables LLMs to generate faithful relational paths through a progressive knowledge distillation strategy. In the online stage, it enhances response reliability via a reasoning-introspection strategy. These strategies effectively explore and leverage structural patterns and constraint priors within knowledge graphs, respectively. Extensive experiments conducted on three benchmark datasets demonstrate that DP achieves new state-of-the-art performance, highlighting its effectiveness, flexibility, and practical applicability. Moreover, the results emphasize the importance of prior exploitation, particularly path generation and constraint extraction, in supporting trustworthy reasoning over knowledge graphs. While this work advances trustworthy LLM reasoning by incorporating knowledge priors, it still relies on human intervention to define constraints when applied to vertical domains. In future work, we plan to investigate automatic methods for extracting and summarizing constraint types, aiming to further reduce manual effort and enhance scalability.

## Acknowledgments and Disclosure of Funding

This work was supported in part by the National Key Research and Development Program of China (2022YFC3303600), the National Natural Science Foundation of China (62306229, 62137002, U22B2019, 62477037, 62293553), the Natural Science Basic Research Program of Shaanxi (2023-JC-YB-593), the Key Research and Development Program of Shaanxi (2024GX-ZDCYL-02-12), the Youth Innovation Team of Shaanxi Universities "Multi-modal Data Mining and Fusion", the Shaanxi Undergraduate and Higher Education Teaching Reform Research Program (23BY195), the Youth Talent Support Program of Shaanxi Science and Technology Association (20240113), the China Postdoctoral Science Foundation (2024M752585, 2025T180425), and the fund of Laboratory for Advanced Computing and Intelligence Engineering.

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

# Technical Appendix

## A    Related Work

**End-to-end Reasoning.** To achieve quick and efficient responses, several studies [31, 33, 34] directly feed questions along with retrieved triples from Knowledge Graphs (KGs) into text decoders. These approaches primarily focus on extracting critical knowledge to answer the questions. One line of research [33, 35, 36] retrieves relevant facts by performing exact entity matching between the input question and the knowledge graph. The retrieved facts are then provided to Large Language Models (LLMs) through prompt engineering to generate answers. To support more semantically rich queries, another line of work [37, 28, 38–40] incorporates knowledge facts into pre-trained models inspired by masked language modeling and subsequently fine-tunes the retriever within the KGQA task. However, such methods often retrieve only the facts associated with the explicit information present in the question, overlooking implicit cues. To address this limitation, some studies [41, 42, 9] propose decomposing complex questions into sub-questions and retrieving knowledge based on semantic similarity between the sub-questions and triples. Nevertheless, these methods often struggle with complex questions, particularly when a large number of retrieved facts are directly input into the text decoder, which can overwhelm the model and impair answer quality. In contrast, our proposed framework, DP, employs the reasoning-introspection strategy to perform path selection and backtracking, effectively avoiding the introduction of false positive paths. It should be noted that our work is the first to employ constraint priors in reasoning to the best of our knowledge.

**Chain-of-thought Reasoning.** With the remarkable success of chain-of-thought prompting, researchers [14, 43, 44] have extended step-by-step reasoning to knowledge graph-based tasks. A prevalent approach [13, 22, 45] first identifies the topic entity in the question, then iteratively retrieves and refines a reasoning path until sufficient factual knowledge is gathered or the answer is obtained, and finally leverages LLMs for answer generation. During the path refinement phase, methods such as DoG [15] utilize in-context learning and multi-agent debate to improve the reliability of generated answers. In contrast to this line of work that decomposes questions step-by-step, another research direction [29, 46, 16] encourages LLMs to directly identify sub-goals of questions and perform reasoning over retrieved knowledge triples. However, both paradigms may produce unfaithful answers, which is a critical issue in high-stakes domains like legal decision-making or medical diagnosis. To mitigate this limitation, recent methods [8, 30] first prompt or fine-tune LLMs to generate relation paths according to questions and then retrieve relevant knowledge triples based on those paths to ground the final answer more faithfully. Unlike previous approaches [8, 30], DP adopts a progressive knowledge distillation strategy to fine-tune LLMs, enabling them to generate more faithful relation paths by more effectively exploiting structural information. This enhanced exploitation is achieved through a more comprehensive collection of weak supervision signals and a preference-aware path generation.

## B    Experiment

### B.1    Experimental Setting

**Training Detail.** The path generator LLaMA3.1-8B-Instruct is loaded from Hugging Face[1] and trained using the LLaMA-Factory[2] framework. All training is conducted on two NVIDIA A800-80GB GPUs, with bfloat16 precision enabled to reduce memory usage and accelerate training. We employ Low-Rank Adaptation (LoRA) to perform an efficient adaptation of large-scale parameters in SFT and KTO. The LoRA configuration uses a rank of 16, an alpha of 32, and a dropout rate of 0.1 and applies to the query, key, value, and output of the self-attention layers. The initial learning rate of SFT training is set to 5e-5, with a warm-up ratio of 0.1. The KTO training uses an initial learning rate of 1e-5, with a preference beta of 0.1 and a warm-up ratio of 0.1. We use the default weights for positive and negative samples, setting $\lambda_p = \lambda_n = 1$. The batch size of SFT and KTO training is set to 4. The datset statistics are shown in Table 7.

**Baseline Introduction.** We compare existing state-of-the-art SL, ICL, and HL-based methods with DP to verify the effectiveness and superiority.

---

[1] https://huggingface.co
[2] https://github.com/hiyouga/LLaMA-Factory

Table 7: The statistics of WebQSP, CWQ, and MetaQA.

| Dataset | KG | Entities | Relations | Triples | Train | Test | Hop |
|---------|-----|----------|-----------|---------|-------|------|-----|
| WebQSP | Freebase | 2,566,291 | 7,058 | 8,309,105 | 2,826 | 1,628 | 2 |
| CWQ | Freebase | 2,566,291 | 7,058 | 8,309,105 | 27,639 | 3,531 | 4 |
| MetaQA | Wiki-Movie | 43,234 | 9 | 133,592 | 329,282 | 30,903 | 3 |

The SL-based methods are introduced as follows. (1) EmbedKGQA [26] addresses multi-hop question answering over sparse KGs by leveraging KG embeddings to predict missing links, effectively mitigating KG incompleteness. Unlike prior approaches, it relaxes answer selection constraints, significantly improving multi-hop reasoning performance on sparse KGs. (2) TransferNet [27] is a transparent and effective model for multi-hop question answering that unifies reasoning over labeled KG relations and textual relations. It iteratively attends to question components and transfers entity scores along activated relations in a differentiable manner, achieving high improvements with interpretable intermediate results. (3) UniKGQA [28] unifies retrieval and reasoning for multi-hop KGQA through a shared model architecture and joint parameter learning. By combining semantic matching with information propagation over KGs, it jointly optimizes retrieval and reasoning, improving accuracy and efficiency on complex questions. (4) RoG [8] integrates LLMs with KGs via a planning-retrieval-reasoning framework that generates KG-grounded relation paths as faithful plans to guide reasoning. This approach enhances reasoning accuracy and interpretability by leveraging structural KG information and supports flexible integration with diverse LLMs. Different from RoG, our proposed framework DP collects question-to-path mappings in the one-to-many form rather than one-to-one. (5) AMAR [10] is an adaptive multi-aspect retrieval framework that enhances LLM reasoning by retrieving and embedding entities, relations, and subgraphs from KGs. It incorporates self-alignment and relevance gating modules to reduce noise and selectively integrate pertinent knowledge, significantly improving accuracy and logical form generation in KGQA tasks. (6) GNN-RAG [24] integrates the graph reasoning capabilities of Graph Neural Networks (GNNs) with the language understanding of LLMs in a retrieval-augmented generation framework. By extracting dense subgraph reasoning paths via GNNs and verbalizing them for LLM reasoning, it effectively addresses multi-hop KGQA and achieves state-of-the-art performance with efficient model sizes.

The ICL-based methods are described below. (1) PoG [29] presents a self-correcting, adaptive planning framework for KG-augmented LLMs that decomposes questions into sub-goals and iteratively explores, updates, and refines reasoning paths over KGs. By integrating guidance, memory, and reflection mechanisms, PoG improves reasoning efficiency and accuracy in complex KGQA tasks. (2) ToG [22] proposes an interactive LLM-KG integration paradigm in which the LLM acts as an agent performing beam search over KGs to identify and reason along promising paths. This training-free, plug-and-play method improves reasoning, knowledge traceability, and correctability, achieving good results with smaller LLMs and reduced computational cost. (3) Readi [30] enables LLMs to efficiently perform multi-hop reasoning over structured data by generating and iteratively editing reasoning paths only when needed. It improves reasoning accuracy and faithfulness while minimizing unnecessary edits, outperforming prior LLM-based methods on both KGQA and TableQA benchmarks. (4) DoG [15] is an iterative interactive KGQA framework that improves LLM reasoning through a subgraph-focusing mechanism and a multi-role debate strategy. By reducing distractions from long reasoning paths and mitigating false-positive relations, DoG enables more accurate and reliable answers in complex KGQA scenarios.

The HL-based methods are summarized as follows. (1) Interactive-KBQA [31] enables LLMs to generate executable logical forms through direct interaction with knowledge bases under minimal supervision. By introducing general APIs and few-shot exemplars for complex questions, it supports step-by-step reasoning and iterative refinement, achieving strong results in low-resource KGQA settings. (2) LightPROF [32] is a lightweight and efficient framework for KGQA that enhances LLM reasoning by structurally integrating KGs into prompts. It retrieves relevant subgraphs, encodes their factual and structural information via a trainable Knowledge adapter, and maps them into the LLM embedding space, enabling effective reasoning with minimal parameter updates.

## B.2 Instruction and Exemplar

We show the instructions and exemplars utilized in the module within the *Planning* and *Introspection* stages.

### B.2.1 Path Generation

```
Please generate relation paths that can help in reasoning to answer the question.
The relation paths must start from the topic entities mentioned in the question. The
 question is: {question}, and the topic entities are: {topic_entities}
```

### B.2.2 Constraint Extraction

```
You will be given a question. Your task is to identify and extract any constraints
present in the question.

**Types of constraints to extract:**

1. Type Constraint:
   - The answer should be of a specific type or category.

2. Multi-Entity Constraint:
   - The question contains multiple entities and requires the answer to
   simultaneously satisfy conditions related to these entities.

3. Explicit Time Constraint:
   - A specific time period or date is mentioned explicitly.

4. Implicit Time Constraint:
   - A time period or date is indirectly implied.

5. Order Constraint:
   - The question involves a sequence or ordering.

Instructions:
- **Extract only present constraints**: Do not add constraints not explicitly or
implicitly mentioned in the question.
- **Output Format**: Return a **List object** with the identified constraints. Each
constraint type should either contain the relevant information or an empty string if
 not applicable.

In-Context Few-Shot
Example 1:
- Question: What country bordering France contains an airport that serves Nijmegen?
- Output: ["1. The answer should be a country", "2. The country borders France", "3.
 The country contains an airport that serves Nijmegen."]

Example 2:
- Question: what did james k polk do before he was president
- Output: ["1. The question implies the time before James K. Polk was president"]

Example 3:
- Question: Who was the 1996 coach of the team owned by Jerry Jones?
- Output: ["1. The team is owned by Jerry Jones", "2. The person was the coach in
1996"]

Example 4:
- Question: What was the last World Series won by the team whose mascot is Lou Seal?
- Output: ["1. The team is the one whose mascot is Lou Seal", "2. The answer should
be a World Series", "3. The World Series is the last one won by the team"]

Example 5:
- Question: What genre is the movie Titanic?
- Output: ["1. The answer should be a genre of the movie"]
```

```
#####
Input question: {question}
Output:
```

### B.2.3  Path Selection

```
Based on the reasoning relation paths in Freebase, think step by step to select the
most one relevant path to answer the question.
You will be given:
- Question: The question to be answered.
- Topic Entities: The main entities identified in the question.
- Memory: Paths have been selected and the feedback of the previous step.
- Reasoning Paths: A set of reasoning paths starting from the topic entities.

In-Context Few-Shot
Example 1:
- Question: What sports team owned by George Steinbrenner did Deion Sanders play
baseball for?,
- Topic Entities: ['Baseball', 'Deion Sanders', 'George Steinbrenner'],
- Memory: [],
- Reasoning Paths: ['Path 1: Baseball -> base.sportbase.sport.played_by_clubs ->
Unknown Entity', 'Path 2: Deion Sanders -> sports.pro_athlete.teams -> Unknown
Entity -> sports.sports_team_roster.team -> Unknown Entity', 'Path 3: Deion Sanders
-> baseball.baseball_player.batting_stats -> Unknown Entity -> baseball.
batting_statistics.team -> Unknown Entity', 'Path 4: George Steinbrenner -> sports.
sports_team_owner.teams_owned -> Unknown Entity']

- Output: {{Path 2}} - This path starts from Deion Sanders, follows his professional
 athlete teams, and connects to a specific sports team. Since the question asks for
the team he played baseball for, this path is the most relevant in identifying the
correct team.

Example 2:
- Question: What movie was Charlie Hunnam in that was about human extinction?,
- Topic Entities: ['Human extinction', 'Charlie Hunnam'],
- Memory: [{{'selected_path': 'Charlie Hunnam -> film.actor.film -> Unknown Entity
-> film.performance.film -> Unknown Entity', 'feedback': 'This path connects Charlie
 Hunnam to films he has acted in, but we need to find which movie is about human
extinction.'}}],
- Reasoning Paths: ['Path 1: Human extinction -> film.film_subject.films -> Unknown
Entity', 'Path 2: Charlie Hunnam -> film.actor.film -> Unknown Entity -> film.
performance.film -> Unknown Entity', 'Path 3: Charlie Hunnam -> common.topic.
notable_types -> Unknown Entity', 'Path 4: Charlie Hunnam -> tv.tv_actor.
starring_roles -> Unknown Entity -> tv.regular_tv_appearance.character -> Unknown
Entity']

- Output: {{Path 1}} - The previous selected path connected Charlie Hunnam to films
he acted in but did not ensure the movie was about human extinction. This path
directly connects "Human extinction" to relevant films, making it the best choice to
 identify the correct movie.

Example 3:
- Question: Which state with Colorado River that Larry Owens was born in?
- Topic Entities: ['Colorado River', 'Larry Owens'],
- Memory: [{{'selected_path': 'Colorado River -> location.location.
partially_containedby -> Unknown Entity', 'feedback': 'This path connects Colorado
River to a location, but we need to find the state Larry Owens was born in.'}}],
- Reasoning Paths: ['Path 1: Larry Owens -> people.person.spouse_s -> Unknown Entity
 -> people.sibling_relationship.sibling -> Unknown Entity']

- Output: {{no path}} - None of the available paths lead to information about the
state where Larry Owens was born.
```

## B.2.4 Constraint Verification

```
Given a question and the associated retrieved knowledge triplets from Freebase, your
 task is to answer the question with these triplets and your knowledge.
You will be given:
- Question: The question to be answered.
- Topic Entity: The main entity identified in the question.
- Constraints: The constraints extracted from the question that should be verified.
- Reasoning Paths: Paths starting from the topic entity (contains only relations).
- Knowledge Triplets: The instantiated reasoning paths in the form of triplets (
entity, relation, entity).

Think step by step to answer the question:
- List all potential answers (**tail entities**) based on the knowledge triplets,
ranking them by how likely they satisfy the question and constraints, and placing
the most likely ones first.
- Check if the answer satisfies the constraints and provide an explanation detailing
 the reasoning process and any missing knowledge needed for full verification.
Return a **JSON object** with the identified constraints.

Important:
- DO NOT output anything except the JSON result.
- DO NOT add explanations, headers, or markdown formatting.
- Return only the JSON object shown in examples.
- Output must be a valid JSON object. All strings and keys must be enclosed in
double quotes.
- Only return the JSON object, no explanation or prefix like "Output:".

Format of the output:
{{"answer": [...], "sufficient": "Yes"/"No", "reason": "..."}}

In-Context Few-Shot
Example 1:
- Question: what is the name of justin bieber brother
- Topic Entity: ["Justin Bieber"],
- Constraints: [],
- Reasoning Path: ["Justin Bieber -> people.person.sibling_s -> Unknown Entity ->
people.sibling_relationship.sibling -> Unknown Entity"],
- Knowledge Triplets: [[["Justin Bieber", "people.person.sibling_s", "m.0gxnnwp"],
["m.0gxnnwp", "people.sibling_relationship.sibling", "Jaxon Bieber"]]]

- Output: {{"answer": ["Jaxon Bieber"], "sufficient": "Yes", "reason": "Based on the
 reasoning path, the answer is Jaxon Bieber, which is the sibling of Justin Bieber
."}}

Example 2:
- Question: What movie was Charlie Hunnam in that was about human extinction?,
- Topic Entities: ["Human extinction", "Charlie Hunnam"],
- Constraints: ["1. The movie is a film Charlie Hunnam acted in.", "2. The movie is
about human extinction."],
- Reasoning Path: ["Charlie Hunnam -> film.actor.film -> Unknown Entity -> film.
performance.film -> Unknown Entity"],
- Knowledge Triplets: [[[["Charlie Hunnam", "film.actor.film", "m.0jwksr"], ["m.0
jwksr", "film.performance.film", "Cold Mountain"]],[["Charlie Hunnam", "film.actor.
film", "m.0jy_sj"], ["m.0jy_sj", "film.performance.film", "Green Street"]],[["
Charlie Hunnam", "film.actor.film", "m.046168c"], ["m.046168c", "film.performance.
film", "Children of Men"]]]]

- Output: {{"answer": ["Children of Men", "Green Street", "Cold Mountain"], "
sufficient": "No", "reason": "The reasoning path connects Charlie Hunnam to films he
 has acted in, but we need to find which movie is about human extinction."}}
```

## B.3 DP performance with various LLMs

The detailed DP performance with various LLMs, presented as mean values with standard deviations, on WebQSP and CWQ is summarized in Table 8.

Table 8: DP performance (mean with standard deviation) on WebQSP and CWQ. The results are averaged over three independent evaluations. We highlight the best performance in bold and the second-best in underline. `LM` denotes the language model used.

| LM | WebQSP | | | CWQ | | | MetaQA | | |
|---|---|---|---|---|---|---|---|---|---|
| | **H** | **H@1** | **F1** | **H** | **H@1** | **F1** | **H** | **H@1** | **F1** |
| LLaMA3.1-8B | $87.9_{\pm 0.2}$ | $82.8_{\pm 0.4}$ | $75.7_{\pm 0.8}$ | $70.8_{\pm 0.3}$ | $61.1_{\pm 0.4}$ | $58.5_{\pm 0.9}$ | $90.2_{\pm 0.1}$ | $87.4_{\pm 0.2}$ | $84.1_{\pm 0.5}$ |
| Qwen3-8B | $75.1_{\pm 0.5}$ | $82.7_{\pm 0.3}$ | $72.3_{\pm 0.6}$ | $76.1_{\pm 0.6}$ | $61.7_{\pm 0.4}$ | $60.2_{\pm 0.7}$ | $90.4_{\pm 0.3}$ | $89.2_{\pm 0.2}$ | $88.6_{\pm 0.2}$ |
| GPT-3.5 | $89.7_{\pm 0.6}$ | $\underline{86.9}_{\pm 0.3}$ | $79.2_{\pm 0.3}$ | $80.0_{\pm 0.6}$ | $72.6_{\pm 0.1}$ | $69.2_{\pm 0.4}$ | $\underline{96.7}_{\pm 0.1}$ | $\underline{95.4}_{\pm 0.3}$ | $90.8_{\pm 0.4}$ |
| GPT-4o | $\mathbf{90.7}_{\pm 0.6}$ | $\mathbf{87.5}_{\pm 0.8}$ | $\mathbf{81.4}_{\pm 0.5}$ | $85.2_{\pm 0.4}$ | $\underline{74.6}_{\pm 0.5}$ | $\mathbf{70.5}_{\pm 0.5}$ | $96.5_{\pm 0.6}$ | $95.2_{\pm 0.4}$ | $94.4_{\pm 0.4}$ |
| GPT-4.1 | $\underline{90.6}_{\pm 0.5}$ | $86.7_{\pm 0.4}$ | $\underline{80.1}_{\pm 0.8}$ | $\mathbf{87.2}_{\pm 0.2}$ | $\mathbf{75.8}_{\pm 0.7}$ | $69.4_{\pm 0.9}$ | $\mathbf{96.8}_{\pm 0.2}$ | $\mathbf{95.5}_{\pm 0.3}$ | $\mathbf{94.9}_{\pm 0.1}$ |

## B.4 Exemplar Impacts

We conduct experiments on CWQ and WebQSP to investigate how the number of exemplars influences response generation. As shown in Figure 5, the performance, measured by H@1, declines as the number of exemplars increases. This degradation may stem from two main factors: (1) A larger number of exemplars results in a longer input context, making it more challenging for the model to capture semantic information effectively. (2) The diversity of exemplars remains relatively unchanged despite the increased quantity, which may cause LLMs to become less confident when encountering out-of-distribution scenarios.

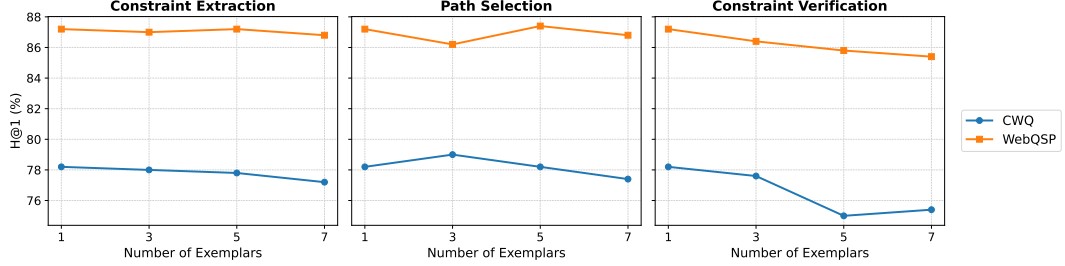

Figure 5: Impact of the number of exemplars on response generation. The results are evaluated by H@1.

## B.5 Interaction with Different LLMs

Figure 6 presents the token consumption and LLM invocation statistics of DP when interacting with various LLMs on the CWQ and WebQSP datasets. The results are averaged over three runs. We see that DP with various LLMs consumes similar tokens and LLM calls to answer a question, except for Qwen3-8B, which shows higher consumption due to its inherently slow thinking mechanisms. It is worth noting that the token consumption for path generation is excluded from this analysis, as that component is executed offline. Overall, these results further validate the practicality and efficiency of the proposed DP framework in real-world scenarios.

## B.6 Case Study

We conduct case studies to qualitatively analyze the strengths and limitations of the proposed framework DP. Figure 7 and 8 illustrate successful cases, while Figure 9 and 10 showcase failure cases. In the successful examples, DP effectively guides LLMs to generate accurate answers by selecting appropriate reasoning paths. When a reasoning path fails to satisfy certain extracted constraints, DP can provide informative feedback, explicitly indicating which constraint is violated.

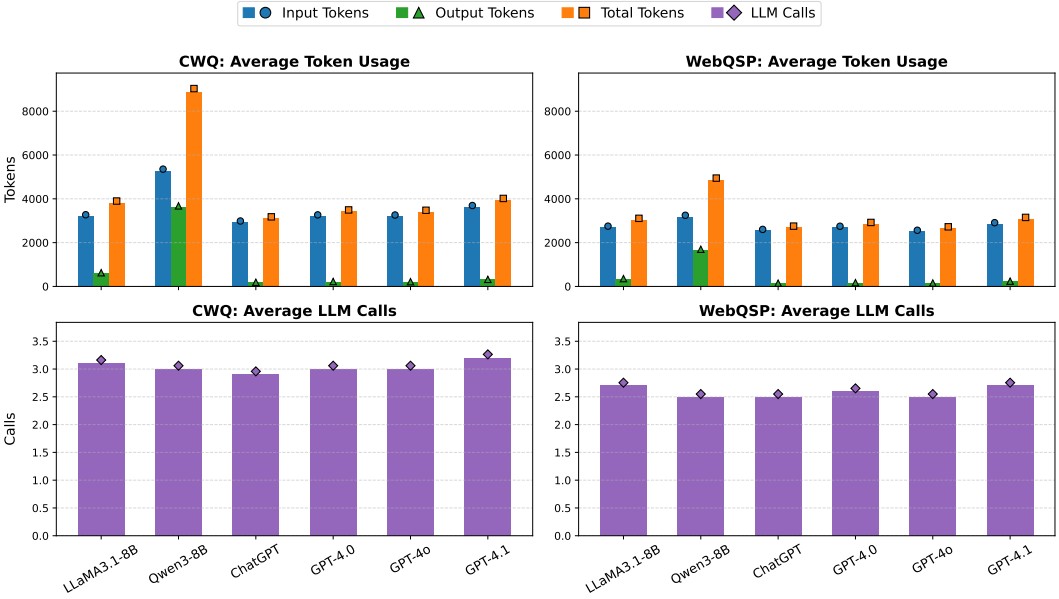

Figure 6: Average token usage and calls of different LLMs.

In contrast, the failure cases highlight two types of errors: incorrect path generation and flawed reasoning. Notably, in the case of path generation errors, the generated paths by DP closely resemble the ground-truth paths, implying that a more robust mechanism may be required better to capture the structural patterns inherent in KGs. In the case of reasoning errors, we see that the LLM fails to cover all accurate answers, although the reasoning paths contain the required knowledge facts.

Question: [Mary] is a notable figure in which religion that is observed in [Indonesia]?

Path Generation:
Path 1: Indonesia -> location.statistical_region.religions ->Unknown Entity -> location.religion_percentage.religion -> Unknown Entity
Path 2: Mary -> people.person.religion -> Unknown Entity
Path 3: Mary -> base.saints.saint.venerated_in -> Unknown Entity

Constraint Extraction:
1. The answer should be a religion
2. The religion is observed in Indonesia
3. Mary is a notable figure in the religion

Path Selection:
{Path 1} - This path starts from Indonesia and connects to religions observed in the country. Since the question is asking about the religion that Mary is a notable figure in that is observed in Indonesia, this path is the most relevant to identify the correct religion.

Instantiation:
[[[['Indonesia', 'location.statistical_region.religions', 'm.03xf2_w'], ['m.03xf2_w', 'location.religion_percentage.religion', 'Islam']],
[['Indonesia', 'location.statistical_region.religions', 'm.03xf301'], ['m.03xf301', 'location.religion_percentage.religion', 'Protestantism']],
[['Indonesia', 'location.statistical_region.religions', 'm.064szk2'], ['m.064szk2', 'location.religion_percentage.religion', 'Catholicism']]]]

Constraint Verification:
{ "answer": ["Islam", "Protestantism", "Catholicism"],
  "sufficient": "No",
  "reason": "The reasoning path connects Indonesia to religions observed in the country. However, we need to further verify which religion Mary, as a notable figure, is associated with in Indonesia."}
(Constraint 1: ✓ Constraint 2: ✓ Constraint 3: ✗ )
====================================================Feedback====================================================
Path:
Path 1: Mary -> people.person.religion -> Unknown Entity
Path 2: Mary -> base.saints.saint.venerated_in -> Unknown Entity

Path Selection:
{Path 1} - This path starts from Mary and connects to her religion. Since the question specifically asks about the religion associated with Mary as a notable figure, this path directly provides the required information.

Instantiation:
[[[['Indonesia', 'location.statistical_region.religions', 'm.03xf2_w'], ['m.03xf2_w', 'location.religion_percentage.religion', 'Islam']],
[['Indonesia', 'location.statistical_region.religions', 'm.03xf301'], ['m.03xf301', 'location.religion_percentage.religion', 'Protestantism']],
[['Indonesia', 'location.statistical_region.religions', 'm.064szk2'], ['m.064szk2', 'location.religion_percentage.religion', 'Catholicism']],
[[['Mary', 'people.person.religion', 'Judaism']], [['Mary', 'people.person.religion', 'Catholicism']]]]

Constraint Verification:
{ "answer": ["Catholicism"],
  "sufficient": "Yes",
  "reason": "Based on the reasoning paths, we find that Indonesia observes Islam, Protestantism, and Catholicism. Mary is associated with both Judaism and Catholicism. Considering the constraints, the answer is Catholicism, which is a religion observed in Indonesia and Mary, a notable figure, is associated with it."}
(Constraint 1: ✓ Constraint 2: ✓ Constraint 3: ✓ )

Answer: "Catholicism"

Figure 7: Successful case requiring path selection again.

Question: What [German Language] speaking European country is also home to the state of [Tyrol]?

🤖 Path Generation:
Path 1: Tyrol -> location.administrative_division.country -> Unknown Entity
Path 2: Tyrol -> location.location.containedby -> Unknown Entity
Path 3: Tyrol -> base.aareas.schema.administrative_area.administrative_parent -> Unknown Entity
Path 4: Tyrol -> location.administrative_division.first_level_division_of -> Unknown Entity
Path 5: German Language -> language.human_language.countries_spoken_in -> Unknown Entity

🧑‍🔧 Constraint Extraction:
1. The country is a European country
2. The country speaks Germanic languages
3. The country is home to the state of Tyrol

🤖 Path Selection:
{Path 1} - This path starts from "Tyrol" and leads to the country it is located in, which directly helps identify the European country that is home to the state of Tyrol. The question is asking for a country (that is in Europe, is home to Tyrol, and where Germanic languages are spoken); since Tyrol is a state/province, following the path from Tyrol to its country is the most direct and relevant way to answer. Path 5 may help identify Germanic language-speaking countries, but combining the two criteria, starting from Tyrol to its country is primary; you can then check if that country is Germanic language-speaking.

🔗 Instantiation:
[[[['Tyrol', 'location.administrative_division.country', 'Austria']]]]

🤖 Constraint Verification:
{ "answer": ["Austria"],
  "sufficient": "Yes",
  "reason": "The reasoning path directly connects Tyrol to Austria as its country. Austria is a European country and, according to general knowledge, German is the official language of Austria. Therefore, Austria satisfies all the constraints: it is a European country, speaks a Germanic language, and is home to the state of Tyrol."}
(Constraint 1: ✅ Constraint 2: ✅ Constraint 3: ✅ )

Answer: "Austria"

Figure 8: Successful case without another iteration.

Question: Before the Euro, what was the currency where [Louis-Eugene Cavaignac] was appointed to governmental position?

🤖 Path Generation:
Path 1: Louis-Eugène Cavaignac -> people.person.nationality -> Unknown Entity -> location.country.currency_used -> Unknown Entity
[Ground Relation Path: [people.person.nationality -> location.country. currency _formerly_used]

🧑‍🔧 Constraint Extraction:
1. The answer should be a currency
2. The currency was in use before the Euro
3. The currency was used in the place where Louis-Eugene Cavaignac was appointed to governmental position
[Only one path, no need to select]

🔗 Instantiation:
[[[['Louis-Eugène Cavaignac', 'people.person.nationality', 'France'], ['France', 'location.country.currency_used', 'CFP franc']],
[['Louis-Eugène Cavaignac', 'people.person.nationality', 'France'], ['France', 'location.country.currency_used', 'Euro']]]]

🤖 Constraint Verification:
{ "answer": ["CFP franc", "Euro"],
  "sufficient": "No",
  "reason": "The reasoning links Cavaignac to France and lists CFP franc and Euro as currencies. However, the Euro is excluded due to timing, and the CFP franc is mainly used in overseas territories, not mainland France where Cavaignac served. Thus, neither currency fully satisfies the constraints."}
(Constraint 1: ✅ Constraint 2: ❌ Constraint 3: ❌ )
[No remaining path, end iteration]

Answer: "CFP franc", "Euro"
Ground truth of the question: ['Assignat', 'French franc']

Figure 9: Case of path generation error.

Question: Which Super Bowl was won by the team that also won the AFC Championship Game in 2013?

🖥 Path Generation

Path 1: Super bowl -> freebase.type_profile.equivalent_topic -> Unknown Entity ->time.recurring_event.instances -> Unknown Entity

Path 2: Super bowl -> freebase.type_profile.equivalent_topic -> Unknown Entity -> sports.sports_championship.events -> Unknown Entity

Path 3: 2013 AFC Championship Game -> sports.sports_championship_event.championship -> Unknown Entity -> time.recurring_event.instances -> Unknown Entity

Path 4:m2013 AFC Championship Game -> sports.sports_championship_event.championship -> Unknown Entity -> sports.sports_championship.events -> Unknown Entity

Path 5:m2013 AFC Championship Game -> sports.sports_championship_event.champion -> Unknown Entity -> sports.sports_team.championships -> Unknown Entity

Path 6: 2013 AFC Championship Game -> sports.sports_championship_event.season -> Unknown Entity -> sports.sports_league_season.championship -> Unknown Entity

🗜 Constraint Extraction:
1. The answer should be a Super Bowl
2. The team won both the Super Bowl and the AFC Championship Game in 2013

🖥 Path Selection:
{Path 5} - This path starts from the 2013 AFC Championship Game, follows to the champion of the event, and then connects that team to their championships. Since the question asks which Super Bowl was won by the team that won the 2013 AFC Championship Game, this path is the most relevant to identify the Super Bowl won by that specific team.

🔗 Instantiation:
[[[['2013 AFC Championship Game', 'sports.sports_championship_event.champion', 'Baltimore Ravens'], ['Baltimore Ravens', 'sports.sports_team.championships', 'Super Bowl XLVII']],
[['2013 AFC Championship Game', 'sports.sports_championship_event.champion', 'Baltimore Ravens'], ['Baltimore Ravens', 'sports.sports_team.championships', '2001 AFC Championship Game']],
[['2013 AFC Championship Game', 'sports.sports_championship_event.champion', 'Baltimore Ravens'], ['Baltimore Ravens', 'sports.sports_team.championships', 'Super Bowl XXXV']]]]}

🖥 Constraint Verification:
{"answer": ["Super Bowl XLVII"],
 "sufficient": "Yes",
 "reason": "According to the triplets, the Baltimore Ravens were the champions of the 2013 AFC Championship Game and have won several championships, including Super Bowl XLVII, Super Bowl XXXV, and the 2001 AFC Championship Game. However, the constraints specify the Super Bowl won by the team that also won the AFC Championship Game in 2013. Only Super Bowl XLVII matches both constraints, as it was the Super Bowl played following the 2012 NFL season, with the AFC Championship Game played in early 2013 and both won by the Baltimore Ravens. The other listed championships do not satisfy the year constraint. Therefore, 'Super Bowl XLVII' is the only answer that satisfies both constraints."}
(Constraint 1: ✅ Constraint 2: ✅ Constraint 3: ✅ )

*Ground truth of the question: ['Super Bowl XXXV','Super Bowl XLVII']*

Figure 10: Case of reasoning error.

