# OpenReview forum: "Deliberation on Priors: Trustworthy Reasoning of Large Language Models on Knowledge Graphs"
_NeurIPS.cc/2025/Conference — NeurIPS 2025 poster_

### Official Review · Reviewer_BKJv · 2025-06-30

**Clarity:** 3
**Significance:** 2
**Originality:** 2
**Rating:** 4
**Confidence:** 3

**Summary:**

The manuscript introduces Deliberation on Priors (DP), a reasoning framework for knowledge graph-based retrieval-augmentation generation (KG-RAG) that utilizes priors embedded in knowledge graphs (KGs). DP first fine-tunes an LLM on a given KG and then distills its knowledge using Kahneman-Tversky optimization. The distilled LLM generates relation paths for each entity mentioned in a question. These paths are instantiated and validated using manually-defined constraints extracted from the question. If validated, the path is used by another LLM to generate an answer. Experiments on three KG question-answering (QA) datasets show that DP outperforms existing approaches.

**Questions:**

1. In the knowledge distillation stage, any path from the source entity to the ground truth entity that involves the source entity’s neighbor within k hops is used as ground truth. However, such paths may be irrelevant to the question. Why are these potentially irrelevant paths not filtered before being used for training?
2. When both DP and the baselines are evaluated using the same language model, does DP still achieve state-of-the-art performance?
3. The test sets for each dataset were uniformly sampled by the authors. Are these samples identical to those used to evaluate the baseline models?
4. How well do naïve LLMs perform on these tasks? In line 261, while the authors state that ‘DP can enhance the reasoning reliability of various LLMs,’ the performance of naïve LLMs is not reported, which weakens this claim.

**Ethical Concerns:**

["NO or VERY MINOR ethics concerns only"]

**Final Justification:**

The manuscript initially raised concerns regarding the fairness of its experimental comparisons, particularly due to the use of different test sets. However, this issue has been addressed in the rebuttal. Taking into account the methodology, the authors' response, and the subsequent discussion, I decided to rate the manuscript as "4: borderline accept".

**Limitations:**

The proposed framework, DP, requires fine-tuning on a specific KG. Consequently, applying it to a different KG requires additional fine-tuning, which limits its scalability and applicability across diverse KGs.

**Paper Formatting Concerns:**

This reviewer did not notice any major formatting issues.

**Quality:**

2

**Strengths And Weaknesses:**

**Strengths**
1. The manuscript is clearly written and easy to follow. Figures and tables are used appropriately and effectively to enhance understanding. Additionally, terms and notations are consistently defined before use, contributing to overall clarity.
2. The idea of validating relation paths using constraints extracted from the questions is compelling. These constraints help ensure that the validated paths align with the question's intent, thereby enhancing the precision of the reasoning process.

**Weaknesses**
1. The constraints are manually defined, requiring manual effort for each domain and question type. This heuristic limits the generalizability of the proposed method. Moreover, manually crafted constraints may not be optimal, which could cap the performance of the framework.
2. In the distillation stage, an LLM is fine-tuned on a specific KG and before knowledge distillation. This approach limits the framework’s scalability, as it requires separate fine-tuning for each KG, reducing its applicability across diverse domains.
3. The supervised learning baselines employ different language models, making it difficult to attribute performance differences solely to the framework. To ensure a fair comparison, experiments using the same underlying language model for both DP and the baselines are necessary.

**Minor Issue**
1. Line 97: ‘a instantiation’ should be corrected to ‘an instantiation’.

---

> ### Author Rebuttal · Authors · 2025-07-31
>
> We sincerely thank you for your thoughtful and constructive feedback. We appreciate your recognition of the clarity of our writing, the novelty of validating relation paths through question constraints, and the effectiveness of our experimental results. We also value your insightful concerns regarding the manual definition of constraints, the scalability of our knowledge distillation process, and the fairness of model comparisons. Your comments have helped us better understand the limitations of our current submission and guided several improvements and clarifications in both the revised draft and the responses below.
>
> In what follows, we address each of your questions and concerns in detail. We also provide additional experimental results to support our claims, clarify methodological decisions, and discuss possible directions for improving the generalizability of our approach.
>
> **W2** The primary goal of fine-tuning in the distillation stage is to enhance the **faithfulness** of LLMs in generating relation paths that align with the structural patterns of the target KG. This is critical not only because the subsequent instantiation module directly relies on these generated paths to retrieve and ground entities from the KG, forming complete reasoning trajectories, but also because high faithfulness is particularly indispensable in **specialized domains such as healthcare and legal services**—fields where the accuracy and trustworthiness of responses are paramount, as unreliable information can lead to severe consequences. Without such fine-tuning, while in-context learning (ICL) can indeed prompt LLMs to generate relation paths, their faithfulness is significantly compromised.
>
> The root cause is that existing LLMs, despite broad knowledge, lack sufficient exposure to the specific structural nuances and relation semantics of a given KG. This leads to paths that are semantically plausible but structurally inconsistent with the KG, undermining downstream reliability.
>
> Notably, our fine-tuning is designed to be **cost-efficient**. We employ LoRA during both SFT and KTO, which efficiently adapts large-scale parameters with minimal computational overhead. Furthermore, the training data for fine-tuning is automatically acquired: the weak supervision signals (question-to-path mappings) are extracted by identifying the shortest traversal sequences from topic entities to answer entities in the KG using the Dijkstra algorithm, eliminating the need for manual construction.
>
> Empirically, our fine-tuning strategy yields substantial improvements in path generation quality: Table 4 shows that the Hit rate of our path generation reaches 94.0% after fine-tuning, confirming the enhanced faithfulness of the generated relation paths. This improvement directly translates to better overall performance, as reflected in Table 2, where our method outperforms ICL-based baselines by over 20 percentage points in F1 score across datasets.
>
> **Minor Issue**: Thank you for pointing this out. We will carefully review the manuscript and revise any potential typos.
>
> **A1** Thank you for raising this important concern. We agree that including irrelevant paths during the distillation stage could introduce noise and affect the model’s reasoning quality. In fact, we have explored several filtering strategies during development, including semantic similarity filtering and LLM-based relevance prediction. However, we observed that these approaches tended to over-filter and remove semantically plausible but indirect reasoning paths.
> For instance, in the case of the question `"What is the name of Justin Bieber’s brother?"`, both of the following paths can lead to the correct answer `"Jaxon Bieber"`:
> 1.
> ```text
> ["Justin Bieber", "people.person.sibling_s", "m.0gxnnwp"],
> ["m.0gxnnwp", "people.sibling_relationship.sibling", "Jaxon Bieber"]
> ```
> 2.
> ```text
> ["Justin Bieber", "people.person.parents", "Jeremy Bieber"],
> ["Jeremy Bieber", "people.person.children", "Jaxon Bieber"]
> ```
> While the second path is logically valid and factually correct, it is often mistakenly filtered out due to low semantic overlap with the original question. As a result, these filtering methods, though effective at removing noise, risk eliminating valuable reasoning paths and reducing coverage of the answer space.
>
> Although we ultimately chose not to apply a filtering mechanism during the distillation phase, this decision allows us to retain a diverse set of reasoning paths, including those that are logically valid but semantically distant from the question. We acknowledge that this choice inevitably introduces some irrelevant paths, which may interfere with downstream reasoning.
>
> To address this, our framework incorporates path selection and constraint validation during the introspection stage. These components are designed to identify and promote paths that are both answer-relevant and constraint-compliant, thereby substantially reducing the negative impact of irrelevant paths on final answer generation.
>
>
> **A2** Thank you for raising this important point regarding model comparability.  In our study, the main baselines we compare against are **ToG**, **PoG**, and **DoG**, as these works all follow the iterative reasoning paradigm similar to **DP**. These baselines represent the state-of-the-art in KGQA, and therefore serve as the most appropriate points of comparison for our proposed method.
>
> To ensure fairness, we conducted experiments using **ChatGPT(GPT-3.5)** and **GPT-4** as the backbone LLMs, aligning with the backbone choices in ToG, PoG, and DoG. These results are reported in **Table 2** of our paper.
>
> For clarity, we reproduce the relevant results from Table 2 below to directly compare the performance of DP with ToG, PoG, and DoG under the same backbone LLMs (**W, C, and M denote the WebQSP, CWQ, and MetaQA datasets, respectively.**):
> |Model|Method|H(W)|H@1(W)|F1(W)|H(C)|H@1(C)|F1(C)|H(M)|H@1(M)|F1(M)|
> |:-:|:-:|:-:|:-:|:-:|:-:|:-:|:-:|:-:|:-:|:-:|
> |ChatGPT|ToG|76.2|-|-|58.9|-|-|-|-|-|
> ||PoG|82.0|-|-|63.2|-|-|-|-|-|
> ||DoG|88.6|62.6|54.2|58.2|49.4|56.6|95.4|85.1|87.2|
> ||**DP**|**89.7**|**86.9**|**79.2**|**80.0**|**72.6**|**69.2**|**96.7**|**95.4**|**90.8**|
> |GPT-4|ToG|82.6|-|36.4|69.5|-|31.8|-|-|-|
> ||PoG|87.3|-|-|75.0|-|-|-|-|-|
> ||DoG|**91.0**|65.4|55.6|56.0|41.0|46.4|**98.3**|90.1|93.1|
> ||**DP**|90.4|**86.7**|**81.7**|**85.6**|**74.6**|**71.1**|96.7|**95.4**|**94.8**|
>
>
> **A3** Thank you for highlighting this important issue regarding test set consistency.
> Indeed, we observed that the baseline methods vary in how they select their test sets—some use the full test set, while others use randomly sampled subsets. This inconsistency makes it difficult to directly align test samples across all prior works.
> In our case, due to the high cost of LLM inference, we sampled a fixed number of examples for evaluation:
> - 500 examples each for WebQSP and CWQ,
> - 600 examples for MetaQA.
> To ensure the fairness and reliability of our comparison, we selected DoG—a strong and representative baseline—and re-evaluated it on the exact same sampled test sets used for DP. This allows for a controlled, head-to-head evaluation using identical data.
> We present the comparison results below:
>
> |Model|Method|H(W)|H@1(W)|F1(W)|H(C)|H@1(C)|F1(C)|H(M)|H@1(M)|F1(M)|
> |:-:|:-:|:-:|:-:|:-:|:-:|:-:|:-:|:-:|:-:|:-:|
> |ChatGPT|DoG*|88.6|62.6|54.2|58.2|49.4|56.6|95.4|85.1|87.2|
> ||DoG**|87.2|62.9|54.1|59.1|48.5|57.5|94.2|86.6|86.8|
> ||**DP**|**89.7**|**86.9**|**79.2**|**80.0**|**72.6**|**69.2**|**96.7**|**95.4**|**90.8**|
> |GPT-4|DoG*|**91.0**|65.4|55.6|56.0|41.0|46.4|**98.3**|90.1|93.1|
> ||DoG**|90.8|65.8|54.9|58.1|39.2|46.2|97.8|91.0|93.2|
> ||**DP**|90.4|**86.7**|**81.7**|**85.6**|**74.6**|**71.1**|96.7|**95.4**|**94.8**|
>
> The "*" represents the original results in the paper, and "**" represents the results when using the same sampling set as DP. W, C, and M denote the WebQSP, CWQ, and MetaQA datasets, respectively.
>
> As shown above, DP consistently outperforms DoG across all datasets when evaluated on the same sampled test sets, further validating the effectiveness of our method under controlled conditions.
>
> **A4** Thank you for raising this point. We would like to clarify that the performance of naïve LLMs was evaluated as part of our ablation study, specifically in Table 3, second row. This row corresponds to the setting where GPT-4.1 is used directly for question answering without any retrieval or reasoning augmentation, representing the baseline performance of a naïve LLM on these tasks.
> As shown in Table 3, the naïve GPT-4.1 model performs significantly worse than our framework on both WebQSP and CWQ datasets. This empirical result directly supports our claim that DP can enhance the reasoning reliability of various LLMs (**W and C denote the WebQSP and CWQ datasets, respectively.**).
> |Setting|H(W)|H@1(W)|F1(W)|H(C)|H@1(C)|F1(C)|
> |:-:|:-:|:-:|:-:|:-:|:-:|:-:|
> |**DP(GPT-4.1)**|**90.6**|**86.7**|**80.1**|**87.2**|**75.8**|**69.4**|
> |GPT-4.1|74.0|71.0|54.6|56.0|53.0|48.9|
> |w/o KTO|88.3|84.7|77.3|86.0|74.6|67.3|
> |w/o PT|88.4|84.8|77.6|86.4|75.0|68.4|
> |w/o EPS|90.0|85.8|79.4|86.6|73.8|68.1|
> |w/o RD|88.3|84.8|77.8|87.2|74.1|67.7|
> |w/o Introspection|86.6|82.0|75.7|85.3|70.8|65.2|
> |w/o CPD|87.8|83.4|76.4|86.2|74.4|68.5|
> |w/o feedback|88.0|83.0|76.5|85.4|73.2|67.1|
>
> We appreciate your attention to this aspect and hope this clarification helps demonstrate the effectiveness of our approach more clearly.

---

> > ### Comment · Reviewer_BKJv · 2025-08-04
> >
> > Thanks for the rebuttal. Some of my concerns have been addressed. Regarding the remaining points, I have a few follow-up questions:
> > - Regarding W1: Can the process of defining constraints be automated?
> > - Regarding W2: Could the authors provide the computational cost required for fine-tuning? Such information would help in assessing the efficiency of the fine-tuning process.
> > - Regarding A2 and A3: As mentioned in W3 of my review, it would be helpful to see results for some of the SL methods, e.g., a comparison of the performance when using LLaMA3.1-8B, as the base LLM for all methods.
> > - Regarding A4: Could the authors also provide the results of naive LLMs that are used in this manuscript, e.g., LLaMA3.1-8B, ChatGPT, GPT-4, and GPT-4o?

---

> ### Author Response · Authors · 2025-08-05
>
> We sincerely thank you for your follow-up questions and for acknowledging the improvements in our rebuttal. Below, we provide our responses to each of the remaining points.
>
> **A1**: Yes, this process can be automated. In fact, as shown in **Table 3** of the paper (row **“w/o CPD”**), we have experimented with an automated approach where the constraint types are not manually predefined. Instead, we use a prompt-based method to have the LLM automatically infer constraint types from the input question. While this automatic variant slightly underperforms the version with predefined constraints, it still shows promising results (only an average 2.18% drop on WebQSP/CWQ). This may suggest that the manually defined constraint types align well with the specific characteristics of the benchmark datasets.
>
> What's more, we have also provided a more detailed discussion on automated constraint extraction in our response to **Reviewer iFUY**.
>
> **A2**: Thank you for pointing this out. To clarify, the computational cost for fine-tuning on 2x NVIDIA A800 (80GB) GPUs is: ~3 hours for the SFT stage (3 epochs) and ~2 hours for the KTO stage (1 epoch). Additional training details (e.g., learning rate, batch size, and LoRA config) are provided in **Appendix B.1** (Training Detail).
>
> We will include this information in the revised version to improve transparency.
>
> **A3**: First, we need to point out the core difference in training paradigm between our approach and existing Supervised learning  methods (e.g., RoG and AMAR):
>
> - **SL Methods**: They directly fine-tune the LLM using **gold answer** labels provided in the KGQA datasets.
> - **DP**: It does **does not rely on any answer supervision**. Instead, it uses **only question–relation path** pairs for offline training, guiding the LLM to generate **faithful relation paths** for KG-based retrieval. Importantly, in the online inference stage, including constraint extraction, path selection and constraint verification, **no fine-tuning is applied**—all steps are carried out by the LLM without additional training.
>
> To ensure a **direct and fair comparison**, we followed your suggestion and aligned our base model with existing  advanced SL methods. To maintain consistency with RoG and AMAR, which utilize LLaMA2-Chat-7B, we carried out additional experiments with DP using **the same model**.
>
> This approach ensures that any performance differences can be primarily attributed to the framework's methodology itself, rather than to the use of a more advanced base model. We believe this is the most equitable comparison scheme under the current circumstances. Results on WebQSP (W) and CWQ (C) are:
>
> |Model|Method|H(W)|H@1(W)|F1(W)|H(C)|H@1(C)|F1(C)|
> |:-:|:-:|:-:|:-:|:-:|:-:|:-:|:-:|
> |LLaMA2-Chat-7B|RoG|85.7|80.8|70.8|62.6|57.8|56.2|
> ||AMAR|84.2|-|**81.2**|-|-|-|
> ||**DP**|**86.1**|**81.3**|71.7|**64.0**|**58.8**|**56.9**|
>
> As these results demonstrate, DP can achieve highly competitive performance **without requiring any answer labels for training**, relying only on relation path supervision.
>
> **A4**: We now include the results of the naive prompting baselines using the LLMs mentioned.
> The table below reports their performance on the WebQSP and CWQ datasets using H@1 and F1 as evaluation metrics, under naive zero-shot prompting without fine-tuning or additional reasoning modules, which clearly shows that DP brings **substantial performance gains** to naive LLMs, confirming its efficacy.
>
> |          Model          |    H@1(W)    |    F1(W)     |    H@1(C)    |    F1(C)     |
> | :---------------------: | :----------: | :----------: | :----------: | :----------: |
> |     LLaMA2-Chat-7B      |     56.4     |     36.5     |     28.4     |     21.4     |
> | **DP** (LLaMA2-Chat-7B) | 81.3 (+24.9) | 71.7 (+35.2) | 58.8 (+30.4) | 59.6 (+38.2) |
> |      LLaMA-3.1-8B       |     55.5     |     34.8     |     28.1     |     22.4     |
> |  **DP** (LLaMA-3.1-8B)  | 82.8 (+27.3) | 75.7 (+40.9) | 61.1 (+33.0) | 58.5 (+36.1) |
> |         ChatGPT         |     59.3     |     43.5     |     34.7     |     30.2     |
> |    **DP** (ChatGPT)     | 86.9 (+27.6) | 79.2 (+35.7) | 72.6 (+37.9) | 69.2 (+39.0) |
> |          GPT-4          |     67.4     |     49.5     |     50.8     |     46.1     |
> |     **DP** (GPT-4)      | 86.7 (+19.3) | 81.7 (+32.2) | 74.6 (+23.8) | 71.1 (+25.0) |
> |         GPT-4o          |     68.4     |     49.5     |     51.5     |     46.9     |
> |     **DP** (GPT-4o)     | 87.5 (+19.1) | 81.4 (+31.9) | 74.6 (+23.1) | 70.5 (+23.6) |
> |         GPT-4.1         |     71.0     |     54.6     |     53.0     |     48.9     |
> |    **DP** (GPT-4.1)     | 86.7 (+15.7) | 80.1 (+25.5) | 75.8 (+22.8) | 69.4 (+20.5) |

---

> > ### Comment · Reviewer_BKJv · 2025-08-05
> >
> > Thanks for the detailed response. Upon reviewing Table 2 and the table in A3 (in the official comment), I noticed that the results for AMAR on CWQ are missing in A3 (in the official comment), and there is no comparison with GNN-RAG. Could the authors clarify why these results, both of which outperform DP, were omitted? Additionally, could the authors provide comparisons on the same test set, similar to the setup in A3 (in the rebuttal)?

---

> ### Author Response · Authors · 2025-08-05
>
> Thank you for your careful review and constructive comments. We address your concerns point by point below:
>
> **Regarding the missing AMAR results on CWQ in A3**: The AMAR method uses LLaMA2-13B for the CWQ dataset and LLaMA2-7B only for WebQSP, as reported in its original paper. Since our method uses LLaMA2-7B, we included AMAR’s WebQSP results for a fair comparison under the same model size. In **Table 2** of our main paper, we see “LLaMA2-7B/13B” are employed in AMAR. Therefore, we omitted the CWQ results from AMAR in table A3 (in the official comment) to avoid comparing across different model sizes.
>
> **Regarding the absence of GNN-RAG**: GNN-RAG is a **hybrid method** that combines graph neural networks (GNNs) with an LLM fine-tuning pipeline. Specifically, it fine-tunes LLaMA2-7B and also leverages **additional GNN-based reasoning components**. To ensure a fair and controlled comparison focused purely on LLM-based reasoning methods, we did not include GNN-RAG in table A3 (in the official comment).
>
> **On test set comparability**: Thank you for pointing this out. Regarding the comparison with SL methods such as RoG and AMAR, we have re-evaluated our method (DP, with LLaMA2-7B) on the **full official test sets** of both **WebQSP** and **CWQ**, which aligns with the evaluation setup used by **RoG** and **AMAR**.
>
> Below we provide the updated performance of DP under this consistent setup. We see that although DP is not trained using the training set, it also achieves SOTA results (evaluated by H). **We need to clarify that DP is a few/zero-shot model from the answer generation view. The fine-tuning and reinforcement learning stages only employ the weak supervision to improve the faithfulness of relation generation. We do not use the answer label to fine-tune LLMs. In contrast, the baselines are fine-tuned using the answer label. We think these can demonstrate the superiority of DP to some extent.**
> |Model|Method|H(W)|H@1(W)|F1(W)|H(C)|H@1(C)|F1(C)|
> |:-:|:-:|:-:|:-:|:-:|:-:|:-:|:-:|
> |LLaMA2-Chat-7B|RoG|85.7|**80.8**|70.8|62.6|57.8|56.2|
> ||AMAR|84.2|-|**81.2**|-|-|-|
> ||**DP**|**86.3**|**80.8**|71.2|**63.7**|**59.0**|**56.8**|
>
> We hope this clarifies the evaluation setup and appreciate your careful attention to the comparability of results.

---

> > ### Comment · Reviewer_BKJv · 2025-08-06
> >
> > Thanks for the response. I trust the authors will incorporate our discussion into the final version of the paper. I have accordingly raised my score.

---

> > > ### Author Response · Authors · 2025-08-06
> > >
> > > Thank you very much for your valuable feedback and the raised score. We greatly appreciate your insightful comments and constructive discussions, which have helped us refine our work significantly. We will carefully incorporate the points from our discussion into the final version of the paper to enhance its quality and clarity.

---

### Official Review · Reviewer_K6r7 · 2025-07-01

**Clarity:** 3
**Significance:** 3
**Originality:** 3
**Rating:** 5
**Confidence:** 4

**Summary:**

This paper proposes "Deliberation on Priors" (DP), a trustworthy reasoning framework for Large Language Models (LLMs) on Knowledge Graphs (KGs). The framework addresses hallucinations in LLMs by better exploiting prior knowledge embedded in KGs, specifically structural information and constraints. DP consists of four modules: (1) Distillation - using progressive knowledge distillation with supervised fine-tuning and Kahneman-Tversky optimization to integrate structural priors; (2) Planning - generating candidate relation paths; (3) Instantiation - grounding paths in KG triples; and (4) Introspection - verifying reasoning paths against extracted constraints with backtracking capability. The authors evaluate DP on three KGQA benchmarks (WebQSP, CWQ, MetaQA) and demonstrate state-of-the-art performance with improved faithfulness and reliability.

**Questions:**

1. Is it feasible to design an automatic constraint extraction strategy to replace the manual predefinition of constraint types? This would further improve the scalability and practicality of your approach in real-world applications, where manual annotation may not be viable.
2. In Section 3.1, you employ Dijkstra's algorithm to identify all shortest relation paths from the subject entity $e_s$ to the ground-truth answer entity $e_t$. In this setting, is there always a unique shortest path for a specific entity pair, or can multiple shortest paths of equal length exist?
3. Table 3 provides a comprehensive ablation study across WebQSP and CWQ datasets, showing the contribution of each component in the DP framework. I notice that removing any of the three modules leads to a similar drop in performance. Could you elaborate on why these three perturbations result in comparable degradation? Does this imply that each component contributes independently and equally to the overall performance?
4. Appendix B.5 (Figure 6) shows the token consumption and LLM invocation statistics for different model integrations. I am curious why integrating DP with **Qwen3-8B** leads to the highest token consumption compared to other LLMs. Is this due to specific characteristics of the model, such as verbosity in generation, or differences in decoding strategies?

**Ethical Concerns:**

["NO or VERY MINOR ethics concerns only"]

**Final Justification:**

The authors have provided a detailed and thoughtful rebuttal that effectively addresses the key concerns raised in my initial review. Specifically, the authors convincingly elaborate on how the current use of predefined constraints can be extended to more diverse domains through a combination of LLM-based in-context induction, semantic parsing to logical forms, and KG schema analysis. Also, I appreciate the authors’ acknowledgment of the limitations when faced with noisy or incomplete KGs. Moreover, their discussion of future extensions (e.g., integrating KG completion and fact verification) shows promising pathways toward deploying the method in real-world settings. So I increased my rating from "borderline accept" to "accept".

**Limitations:**

yes

**Quality:**

3

**Strengths And Weaknesses:**

**Strengths**
1. The challenge addressed in this paper, enhancing the faithfulness and reliability of response generation, is highly relevant to KG-based Retrieval-Augmented Generation (RAG). To this end, the authors introduce a trustworthy reasoning framework, DP, which enables large language models to generate faithful and reliable responses through deliberate reasoning over priors embedded in knowledge graphs.

2. Extensive experiments across three datasets with multiple LLMs (LLaMA, GPT variants) demonstrate the framework's flexibility and consistent improvements. The ablation studies provide good insights into component contributions.

3. The method requires significantly fewer LLM calls (2.5-2.9 vs 10-22 for baselines) while maintaining superior performance, making it practically valuable.

**Weaknesses**
1. One limitation of the proposed method is the manual definition of constraint types. As shown in Table 1, these types are predefined rather than automatically extracted. However, in real-world applications, it is often necessary to extract such constraints automatically. Although the authors include experiments comparing the performance of DP with and without manually defined constraints, the performance of response generation degrades without manual input.

2. Testing on only 500 samples from WebQSP/CWQ due to "computational constraints" raises questions about the robustness of conclusions, especially given the relatively small performance margins in some cases. The shortest-path heuristic for collecting supervision signals may not capture all valid reasoning paths, potentially limiting the diversity of learned patterns. The paper acknowledges this limitation but doesn't thoroughly investigate its impact.

---

> ### Author Rebuttal · Authors · 2025-07-31
>
> We sincerely thank you for your thoughtful and constructive review. We greatly appreciate your recognition of the motivation and contributions of our work—particularly the trustworthiness of the DP framework, the practical value of reduced LLM calls, and the thoroughness of our experiments and ablations.
> Your comments and questions helped us reflect more deeply on both the strengths and limitations of the current design. Below, we address each of your concerns in detail and clarify several points that were unclear in the original version.
>
> **A1** You raise an important point regarding the scalability of our framework when constraint types are manually defined. We fully agree that enabling automatic constraint acquisition would significantly improve the practicality of our approach in real-world applications.
>
> In fact, we conducted an ablation study (see **Table 3, row “w/o CPD”**) where constraints were not manually predefined. Instead, we used a prompt-based method to let the LLM automatically identify constraint types from the input question. While this automatic variant slightly underperforms the version with predefined constraints, it still shows promising results. This may suggest that the manually defined constraint types align well with the specific characteristics of the benchmark datasets. However, their effectiveness may not generalize to more diverse or open-domain settings, where more flexible and adaptive constraint extraction strategies would be required.
>
> In future work, we plan to explore this direction through the following approaches:
> - Leveraging prompt-based methods to guide large language models in analyzing representative questions and automatically identifying, abstracting, and generalizing common constraint patterns within a target domain.
> - Exploiting the schema or ontology of the knowledge graph, which encodes valuable structural and semantic information (e.g., entity types, relation domains and ranges). By analyzing this metadata, the system can infer potential constraints. For example, if a relation’s domain is “Film” and its range is “Year,” any query involving this relation can be automatically linked to a constraint such as “the answer type must be a year.”
>
> **A2** In our setting, multiple shortest paths of equal length may exist between a given subject entity $e_s$ and answer entity $e_t$, and we account for this in our supervision process. For instance, for the question `"What is the name of Justin Bieber’s brother?"`, both of the following reasoning paths can lead to the correct answer `"Jaxon Bieber"`:
>
> ```text
> 1. ["Justin Bieber", "people.person.sibling_s", "m.0gxnnwp"], ["m.0gxnnwp", "people.sibling_relationship.sibling", "Jaxon Bieber"]
> 2. ["Justin Bieber", "people.person.parents", "Jeremy Bieber"], ["Jeremy Bieber", "people.person.children", "Jaxon Bieber"]
> ```
> To capture such diversity, we use Dijkstra’s algorithm to collect all shortest-length relation paths, rather than selecting just one. This helps us maximize the supervision signal and enables the model to learn from multiple semantically equivalent reasoning routes.
>
> **A3** While Table 3 shows that removing any of the three modules—constraint extraction, path selection, or constraint verification—results in a comparable performance drop, this does not imply that these modules contribute independently or equally to the overall performance.
>
> Each of the three modules plays a distinct yet interdependent role in the DP framework:
> - Constraint extraction provides semantic guidance by identifying meaningful conditions to be satisfied.
> - Path selection ensures the reasoning process focuses on plausible paths that align with both the query and the extracted constraints.
> - Constraint verification serves as a filtering mechanism that enforces answer validity by cross-checking constraints post hoc.
>
> Although the degradation appears numerically similar across ablations, this is likely due to the fact that removing any single module breaks the coherence of the pipeline. For instance, without constraint extraction, the subsequent modules lack semantic grounding; without path selection, the search space becomes noisy and inefficient; and without constraint verification, spurious or weakly supported answers may be retained.
>
> In this sense, the performance drops are not indicative of independent or redundant contributions but rather suggest that these components are synergistic—each is critical, and disabling one disrupts the end-to-end reasoning process.
>
> **A4** The higher token consumption observed with Qwen1.5-8B is mainly due to its verbose generation style. Qwen3-8B is optimized for reasoning tasks and tends to explicitly articulate its intermediate reasoning steps, resulting in longer outputs compared to other models like GPT or LLaMA. This verbosity, while potentially beneficial for transparency and interpretability, naturally increases token usage during inference.

---

> > ### Author Response · Authors · 2025-08-05
> > **Brief Follow-up on Rebuttal**
> >
> > Dear Reviewer #K6r7,
> >
> > Thank you again for your positive and encouraging feedback—I truly appreciate it.
> >
> > I just wanted to gently confirm that my responses to your comments were received. Please let me know if any further clarification would be helpful.
> >
> > Best regards,
> >
> > Authors

---

### Official Review · Reviewer_jD5U · 2025-07-02

**Clarity:** 3
**Significance:** 3
**Originality:** 3
**Rating:** 4
**Confidence:** 3

**Summary:**

The paper tackles hallucinations and factual inaccuracies in knowledge graph-based retrieval-augmented generation by introducing Deliberation on Priors (DP). This two-stage framework leverages structural and logical priors from KGs. In the Progressive Knowledge Distillation stage, LLMs are fine-tuned using KG-derived reasoning paths with a combination of supervised learning and Kahneman-Tversky optimization to incorporate structural priors. The subsequent Reasoning-Introspection stage employs a deliberative verification process that utilises symbolic logical constraints to iteratively refine and filter reasoning paths, thereby reducing hallucinations and enhancing output reliability. Experiments on three KGQA benchmarks demonstrate improved performance, supported by ablation studies that validate the contribution of each component. The authors also experiment and report LLM calls and token counts for practical deployment consideration.

**Questions:**

* In L125, shouldn’t the function $M$ also have $G$ as the input? Additionally, it is somewhat confusing to write the function defined over the full question set; perhaps writing it over $q$ itself is better.
* L128, what is the backup option when the reasoning depth is not available?
* A minor suggestion: Eq (1), *P_w (q)* is a set, so maybe as a convention, write it in bold so that it is clear straightaway.
* As with any KG or RAG-based framework, performance is highly dependent on the quality of the retrievals/ KG. Can the DP framework also be used to send feedback to the KG retrieval and improve it?
* SFT and KTO are interesting approaches. In practice, how is this performed? First, SFT is done, and then KTO, or do we keep alternating between them? Was there any ablation performed on this?
* L178: What does the author precisely mean by semantic alignment? How is this calculated?
* L200: Backtracking is an interesting part of the framework, and it would be interesting to see in practice how often, for each dataset, backtracking was triggered.
* Minor: I found Figures 7-9 very interesting and believe there should be discussion or references to them in the main paper as well.
* Table 2:
    - I believe the std-dev should be reported here itself rather than a different table in the appendix.
    - Grouping the methods on the models would be better to see the performance increase.
    - Why is that with ICL, GPT-4o, GPT-4.1, Llama3.1-8B are not experimented with?
    - Minor: WebQSP: Llama2-Chat-7B also has 90.7.
    - In terms of stochasticity in DP, what is the main source of randomness here? Is a higher temperature used in LLMs in the experiments?
* Table 5, isn’t it a bit unfair comparison as DP has a substantial fine-tuning cost associated, which is not reflected here?
* Table 4: Why is the performance close to 99 (H) for CE? If it is uniformly sampled, then it should roughly be close to what is presented in Table 2. What am I missing here?
* In future, what do authors see as the possible methods to incorporate more dynamic constraints and big KGs with not pre-defined reasoning hops?

**Ethical Concerns:**

["NO or VERY MINOR ethics concerns only"]

**Final Justification:**

I will update my score as all my concerns are addressed.

**Limitations:**

Authors briefly discuss the limitations with the conclusions in the paper. Authors can discuss the limitations in detail in the appendix. I do not see any direct negative societal impacts.

**Paper Formatting Concerns:**

No concerns with the paper formatting.

**Quality:**

2

**Strengths And Weaknesses:**

The paper presents a technically solid and methodologically rigorous approach addressing hallucination and faithfulness issues in knowledge graph-augmented large language model reasoning. The authors’ two-stage framework—combining progressive knowledge distillation, which incorporates structural priors through supervised fine-tuning augmented with Kahneman-Tversky optimization, and a reasoning-introspection module that leverages symbolic logical constraints for output verification. This is a well-motivated and novel integration. The experimental evaluation is good, covering multiple KGQA benchmarks with comprehensive ablations that isolate the contributions of each component. The inclusion of practical efficiency metrics, such as LLM call and token usage, demonstrates awareness of deployment constraints, thereby increasing the work’s applicability. The paper effectively reduces hallucinations while improving interpretability, offering a promising direction for integrating structured priors into LLM reasoning.

I see a few weaknesses in the paper related to reliance on constraints in the introspection stage, effectiveness of the KTO optimization step, and experiment with different LLMs, which I discuss in the questions section below.

---

> ### Author Rebuttal · Authors · 2025-07-31
>
> We thank you for your thoughtful and constructive feedback. We appreciate your recognition of this work.
>
> **A1** We agree that the function definition in Line 125 could be made more precise and readable. To improve clarity, we will revise the statement in Line 125 to："**we define a function $\mathcal{M}$ that maps a question $q \in \mathcal{Q}$ to a set of plausible relation paths...**". This change emphasizes that the function operates on individual questions, rather than on the full question set. The complete input to the function $\mathcal{M}$ is clearly specified in Equation 1: $\mathcal{M}(q, e_s, \mathcal{G}, k)$ .
>
> **A2** When the reasoning depth is not available, our framework defaults to a general backup strategy: we perform shortest-path search over the entire knowledge graph. Specifically, the parameter k in our implementation represents the maximum allowed reasoning depth and is typically provided by the dataset. Extracting a k-hop subgraph centered at the topic entity serves to narrow the search space, significantly improving efficiency during path extraction and instantiation. However, in the absence of such a predefined depth, we fall back to querying the full graph for shortest paths between the topic and target entities. While this ensures correctness, it results in increased computational overhead and longer runtimes due to the expanded search space.
>
> **A3**  You are absolutely right that $\mathcal{P}_w(q)$ represents a set, and using boldface notation would improve clarity and consistency. We will revise the notation accordingly in the final version to better reflect its set nature.
>
> **A4**  Yes, the DP framework does include a feedback mechanism that improves the effectiveness of KG retrieval, though not in the traditional sense of updating the retriever module directly.
>
> DP transforms the triple retrieval into relation path generation. Specifically, it first generates multiple candidate relation paths and then instantiates them over the KG to ground the answer. In this architecture, retrieval is guided by structured reasoning paths rather than by query-keyword matching, and path selection becomes the central mechanism of retrieval.
>
> The introspection module plays a critical role in enabling feedback. It evaluates whether the instantiated answers satisfy the pre-extracted constraints. If a candidate path leads to an answer that violates these constraints or lacks sufficient information, the system triggers backtracking to select an alternative relation path from the candidate pool. This mechanism functions as an implicit feedback loop that iteratively refines the path selection process to improve final answer quality.
>
> Looking forward, such feedback could inform KG completion (e.g., when certain paths repeatedly fail to instantiate) or guide refinement of the path generator (e.g., when paths consistently violate constraints).
>
> **A5**  In our framework, the training is conducted in two distinct stages: SFT followed by KTO. We do not alternate between them during training.
> - SFT teaches the model to generate plausible reasoning paths from input questions in a supervised manner, establishing the basic question-to-path mapping.
> - KTO then refines this ability by using preference data to distinguish high-quality paths from suboptimal ones, encouraging the generation of semantically coherent and structurally valid reasoning.
>
> As shown in Table 3, removing KTO leads to a clear performance drop, highlighting its essential role in enhancing the robustness and faithfulness of the overall DP pipeline.
>
> **A6**  We agree that the use of the term semantic alignment was not clearly defined in the original text, and we will revise the description to improve clarity.
>
> What we intended to convey is that semantic alignment refers to the initial relation path selection step, where the model selects the most semantically relevant path to the input question from a set of candidates. At this stage, no verification feedback is available yet—the model relies solely on the semantics of the question and the candidate paths. In subsequent iterations (after backtracking), the model can incorporate feedback signals from the constraint verification module to refine the selection process.
>
> We will clarify this distinction between initial and feedback-informed path selection in the revised version and refer to Appendix B.2.3 for further details.
>
> **A7**  It is indeed one of the core innovations of our framework. Measuring how frequently backtracking is triggered in practice provides useful insight into how often the system encounters and resolves constraint violations during reasoning.
>
> We analyzed our test-time logs and computed the average number of backtracking steps per question for each dataset, summarized below:
> |Dataset|GPT-3.5|GPT-4.1|
> |:-:|:-:|:-:|
> |WebQSP|0.07|0.21|
> |CWQ|0.10|0.42|
>
> GPT-4.1 triggers backtracking more often than GPT-3.5, likely due to its stronger instruction-following ability, which leads to stricter constraint checking. In contrast, GPT-3.5 may rely more on internal heuristics and skip verification steps. This also corresponds with Table 2, where GPT-4.1 yields better performance.
>
> **A8**  In the final version, we will add a dedicated case study section to highlight the significance of these examples, along with relevant references and brief discussions in Sections 3.2 and 3.3 to further illustrate their connection to the strengths and limitations of our framework.
>
> **A9**  We appreciate your detailed suggestions regarding Table 2. Below we address each point in turn:
>
> 1. You are absolutely right that reporting the std-dev directly in Table 2 improves clarity. The reason we placed it in the appendix originally was due to space limitations in the main paper. In our revised version, we will integrate the std-dev values into the main table for better accessibility.
> 2. This is a very helpful suggestion. We have reorganized the table to group methods based on the underlying LLMs, which makes performance comparisons more intuitive. Please see the updated version below (**W, C, and M denote the WebQSP, CWQ, and MetaQA datasets, respectively.**):
> |Model|Method|H(W)|H@1(W)|F1(W)|H(C)|H@1(C)|F1(C)|H(M)|H@1(M)|F1(M)|
> |:-:|:-:|:-:|:-:|:-:|:-:|:-:|:-:|:-:|:-:|:-:|
> |GPT-3.5|ToG|76.2|-|-|58.9|-|-|-|-|-|
> ||PoG|82.0|-|-|63.2|-|-|-|-|-|
> ||DoG|88.6|62.6|54.2|58.2|49.4|56.6|95.4|85.1|87.2|
> ||**DP**|**89.7**|**86.9**|**79.2**|**80.0**|**72.6**|**69.2**|**96.7**|**95.4**|**90.8**|
> |GPT-4|ToG|82.6|-|36.4|69.5|-|31.8|-|-|-|
> ||PoG|87.3|-|-|75.0|-|-|-|-|-|
> ||DoG|**91.0**|65.4|55.6|56.0|41.0|46.4|**98.3**|90.1|93.1|
> ||**DP**|90.4|**86.7**|**81.7**|**85.6**|**74.6**|**71.1**|96.7|**95.4**|**94.8**|
>
> 3. Our main experimental setting uses GPT-3.5 and GPT-4, which are also the backbones used in baseline works like ToG, PoG, and DoG. This ensures a fair comparison under consistent model conditions. The additional experiments with GPT-4o, GPT-4.1, and LLaMA3.1-8B are primarily meant to validate the generalizability and compatibility of DP across a range of modern models, rather than to compare with prior baselines under few-shot ICL.
>
> 4. Thank you for pointing this out—we acknowledge this omission and will correct it in the updated table.
>
> 5. Yes, the DP pipeline does involve a certain level of randomness, mainly due to using a sampling temperature of 0.8 when invoking LLMs during path generation and introspection. However, as shown in Table 7, the std-dev across runs remains below 1.0, indicating that DP produces stable and reliable results despite this inherent stochasticity.
>
> **A10** We agree that DP has fine-tuning costs, but in this analysis, we focus on the costs associated with generation and inference.
>
> The training cost is a one-time cost, whereas the inference cost is incurred repeatedly with each query. Therefore, the total cost for generating responses is a combination of both the training and inference costs. As the number of queries (n) increases, the training cost can be spread over more queries, which makes it less significant in comparison to the growing inference cost.
>
> To clarify this, we use the following formula to express the total cost: $C_{\text{total}} = \frac{C_{\text{train}}}{n} + n \cdot C_{\text{infer}}$.
>
> As $n$ increases, the inference cost becomes more important, and the training cost for each query decreases. This means that as the number of queries increases, inference cost becomes the main factor and becomes increasingly important over time. This gives DP an efficiency advantage in real-world scenarios.
>
> **A11**  We appreciate your attention to the evaluation details. There may be a misunderstanding regarding the hit rate (H) reported for CE in Table 4. Here, CE stands for **Constraint Extraction**, and the H metric measures how accurately the model extracts constraints from questions—not answer correctness. Specifically, we annotated 100 examples with gold-standard constraints and computed how often the model’s output matched them. The high score (~99) reflects strong extraction accuracy on this subset and is not directly comparable to the H scores in Table 2.
>
> **A12** Our current DP framework uses a set of predefined constraints, which work well in controlled settings. However, enabling automatic constraint acquisition is essential for broader applicability. In future work, we plan to:
>
> - Use LLMs to analyze representative questions and generalize common constraint patterns within a domain.
> - Leverage KG schema or ontology (e.g., entity types, relation domains/ranges) to infer likely constraints. For example, if a relation’s range is “Year,” the system can enforce type consistency for answers.
>
> Additionally, as noted in our response to Q2, the fixed reasoning hop limit is used only during training to bound the subgraph for efficient path extraction. The framework does not rely on a fixed number of hops and can naturally scale to large KGs with flexible or unbounded reasoning depth.

---

> > ### Author Response · Authors · 2025-08-05
> > **Gentle Follow-up on Response to Reviewer Questions**
> >
> > Dear Reviewer # jD5U，
> >
> > I hope this message finds you well.
> > I wanted to thank you again for your thoughtful feedback on our submission to NeurIPS.
> >
> > I completely understand that you may be reviewing many papers and have a busy schedule. However, I just wanted to gently follow up in case my responses were overlooked or if there is any additional information I could provide to assist in your evaluation.
> >
> > Thank you again for your time and consideration. I truly appreciate your efforts in helping improve our work.
> >
> > Best regards,
> >
> > Authors

---

> > ### Comment · Reviewer_jD5U · 2025-08-05
> >
> > Thank you for the detailed responses. A quick question: What significance level value is used for the bolding in the table?

---

> > > ### Author Response · Authors · 2025-08-05
> > >
> > > Thanks for your comments. The bolded values indicate the highest numerical values in each column with the same setting. The results of DP are averaged over three independent evaluations. We show the standard deviation in Table 7.

---

> > > > ### Comment · Reviewer_jD5U · 2025-08-07
> > > >
> > > > The bolding should be based on a two-tailed p-test to determine if the differences are statistically significant across runs. Could the authors confirm whether this was done, and if so, what significance level was used? If not, I would request that the authors perform this test and bold the values in the table.

---

> > > > > ### Author Response · Authors · 2025-08-08
> > > > >
> > > > > We sincerely thank the reviewer for the insightful comment.
> > > > >
> > > > > In the original submission, the performance of DP was reported as the **average of three independent runs**, without formal statistical testing. To address your concern, we have conducted **seven additional independent runs**, resulting in a total of **10 runs** for DP. Based on this expanded set of results, we have performed formal statistical significance tests to ensure that any bolded improvements over baselines are not only numerically higher but also statistically meaningful.
> > > > >
> > > > > ### Statistical Testing Methodology
> > > > >
> > > > > The baseline results reported in the table are based on **single-run evaluations**, and therefore we treat each baseline value as a **fixed reference point** (i.e., a known constant) rather than a sampled distribution. Given this setup, for each metric in which DP achieved the best performance, we conducted a **one-sample, one-tailed t-test** to assess whether the mean performance of DP across 10 runs is significantly greater than the corresponding baseline value.
> > > > >
> > > > > We employ a **one-tailed test** because our hypothesis is directional: we aim to determine whether DP **significantly outperforms** the best baseline on each metric, rather than simply differing from it.
> > > > >
> > > > > The hypotheses are formally defined as:
> > > > >
> > > > > - **Null hypothesis ($H_0$):** $μ_{\mathrm{DP}} ≤ \mu_{0}$
> > > > > - **Alternative hypothesis ($H_1$):** $μ_{\mathrm{DP}} > \mu_{0}$
> > > > >
> > > > > where $μ_{\mathrm{DP}}$ represents the true population mean of DP’s performance across multiple runs, and $\mu_0$ denotes the fixed performance value of the best competing method.
> > > > >
> > > > > ### Significance Threshold
> > > > >
> > > > > We adopt the conventional significance level of $\alpha=0.05$. Only those improvements for which the one-tailed t-test yields $p < 0.05$ are considered statistically significant and will be highlighted (bolded) in the final version of the paper.
> > > > >
> > > > > ### Experimental Results
> > > > >
> > > > > Below, we present key metrics where DP outperformed the best baseline. For each, we report the mean and standard deviation of DP’s performance over 10 runs, the fixed baseline value, the source method, and the resulting p-value:
> > > > >
> > > > > | Metric             | DP (mean ± std) | Baseline (fixed) | Baseline Method | p-value  | Significance |
> > > > > |--------------------|------------------|-------------------|------------------|----------|------------------------|
> > > > > | H(W) - GPT-3.5     | 89.8 ± 0.53      | 88.6              | DoG              | <0.001   | ✅                     |
> > > > > | H@1(W) - GPT-3.5   | 86.9 ± 0.41      | 62.6              | DoG              | <0.001   | ✅                     |
> > > > > | F1(W) - GPT-3.5    | 79.2 ± 0.31      | 54.2              | DoG              | <0.001   | ✅                     |
> > > > > | H(C) - GPT-3.5     | 80.1 ± 0.39      | 63.2              | PoG              | <0.001   | ✅                     |
> > > > > | H@1(C) - GPT-3.5   | 72.3 ± 0.51      | 49.4              | DoG              | <0.001   | ✅                     |
> > > > > | F1(C) - GPT-3.5    | 69.0 ± 0.48      | 56.6              | DoG              | <0.001   | ✅                     |
> > > > > | H(M) - GPT-3.5     | 96.5 ± 0.29      | 95.4              | DoG              | <0.001   | ✅                     |
> > > > > | H@1(M) - GPT-3.5   | 95.3 ± 0.31      | 85.1              | DoG              | <0.001   | ✅                     |
> > > > > | F1(M) - GPT-3.5    | 90.8 ± 0.35      | 87.2              | DoG              | <0.001   | ✅                     |
> > > > > | H@1(W) - GPT-4.0   | 86.9 ± 0.40      | 65.4              | DoG              | <0.001   | ✅                     |
> > > > > | F1(W) - GPT-4.0    | 81.8 ± 0.28      | 55.6              | DoG              | <0.001   | ✅                     |
> > > > > | H(C) - GPT-4.0     | 85.7 ± 0.39      | 75.0              | PoG              | <0.001   | ✅                     |
> > > > > | H@1(C) - GPT-4.0   | 74.5 ± 0.51      | 41.0              | DoG              | <0.001   | ✅                     |
> > > > > | F1(C) - GPT-4.0    | 71.0 ± 0.57      | 46.4              | DoG              | <0.001   | ✅                     |
> > > > > | H@1(M) - GPT-4.0   | 95.4 ± 0.28      | 90.1              | DoG              | <0.001   | ✅                     |
> > > > > | F1(M) - GPT-4.0    | 94.7 ± 0.43      | 93.1              | DoG              | <0.001   | ✅                     |
> > > > >
> > > > > As shown, all highlighted improvements achieve **highly significant** p-values (all < 0.001), providing strong statistical evidence that DP consistently outperforms the best baseline methods.
> > > > >
> > > > > ### Revisions to the Manuscript
> > > > >
> > > > > We will make the following updates in the revised paper to enhance clarity and methodological rigor:
> > > > >
> > > > > 1. **Add a clear statement** in the experimental section explaining that bolded results indicate **statistically significant improvements ($p < 0.05$)**, as determined by a **one-sample, one-tailed t-test** comparing DP’s distribution of scores against the fixed baseline value.
> > > > > 2. **Revise all result tables** to ensure that **only statistically significant improvements** are bolded, thereby aligning visual emphasis with statistical evidence.

---

> > > > > > ### Comment · Reviewer_jD5U · 2025-08-08
> > > > > >
> > > > > > Thank you for the response! Yes, please revise the manuscript with statistical tests and bolding. I will update my score as all my concerns are addressed.

---

> > > > > > > ### Author Response · Authors · 2025-08-08
> > > > > > >
> > > > > > > Thank you for your feedback and for raising the score. We truly appreciate your insightful comments and constructive suggestions, which have been instrumental in helping us refine our work. We will carefully address all the points discussed and incorporate them into the final version of the paper to further enhance its quality, clarity, and rigor.

---

### Official Review · Reviewer_iFUY · 2025-07-04

**Clarity:** 4
**Significance:** 3
**Originality:** 3
**Rating:** 4
**Confidence:** 4

**Summary:**

The paper presents a framework called DP to enhance the reasoning capabilities of LLMs when performing tasks such as KGQA. The framework incorporates a progressive knowledge distillation strategy that integrates the structural priors of KGs into LLMs, improving the faithfulness of relation path generation. Furthermore, DP employs a reasoning-introspection strategy, which verifies whether generated relation paths satisfy predefined constraints, thereby improving the reliability of responses. Extensive experiments on several benchmark datasets demonstrate that DP outperforms current methods, with a 13% improvement in Hit@1 on ComplexWebQuestions.

**Questions:**

1. The framework relies on predefined constraints for introspection. Could the authors elaborate on potential methods for automating the extraction of these constraints? How would this work in more diverse domains or with complex, domain-specific knowledge graphs?

2. Can the authors provide further insights into the failure cases where DP struggled to generate reliable paths? This would help in understanding its limitations better, especially in more complex reasoning tasks.

3. While the proposed method performs well on benchmark datasets, how does it handle real-world applications where knowledge graphs may be incomplete or noisy?

**Ethical Concerns:**

["NO or VERY MINOR ethics concerns only"]

**Limitations:**

Yes.

**Quality:**

3

**Strengths And Weaknesses:**

**Strengths:**

The paper provides a thorough methodology for improving LLM reasoning by incorporating priors from knowledge graphs, which is a relevant issue in current LLM research. The proposed framework (DP) is novel and is shown to outperform existing methods on multiple benchmark datasets.

The paper is well-structured and provides clear explanations of the methodology, experimental setup, and results. The figures and examples further aid in understanding the approach and its effectiveness.

The use of a combination of knowledge distillation and reasoning introspection for enhancing reasoning over knowledge graphs is a novel contribution to the field of LLMs and KG-based retrieval-augmented generation.

**Weaknesses:**

1. While the experiments are extensive, the paper could benefit from deeper analysis of the failure cases, especially those that involve complex, multi-hop reasoning where the proposed method might still struggle. While error analysis is presented, more detailed insights into these cases could help improve the robustness of the framework.

2. The framework relies heavily on predefined constraints, which could limit its flexibility. The authors mention plans for automating the extraction of these constraints, but further discussion on how this would work in diverse domains would be valuable.

3. The method relies on high-quality KGs, which limits its usefulness in real-world scenarios.

---

> ### Author Rebuttal · Authors · 2025-07-31
>
> Thanks for your valuable and constructive comments.
>
> **A1** Our DP framework does rely on a set of predefined constraints that are universal and effective. In **Tabel 3**, we conducted an ablation study where constraints were not manually predefined (**row “w/o CPD”**), but instead a prompt-based approach was used to enable LLM to automatically identify constraint types from the input problem. While this automatic variant slightly underperforms the version with predefined constraints, it still shows promising results (the average decrease in the three indicators on WebQSP and CWQ is only **2.18%**). This may suggest that the manually defined constraint types align well with the specific characteristics of the benchmark datasets.
>
> We fully agree that automatically extracting constraints is a crucial step to ensure that the framework generalizes to more diverse and specialized domains. In future work, we plan to explore the following aspects:
>
> - **LLM-based In-Context Induction** This method leverages the powerful instruction-following and pattern-learning capabilities of LLMs. By designing specific prompts and providing a few (few-shot) examples of "question-constraint" pairs, we can guide the LLM to learn and generalize the patterns for identifying corresponding constraint structures in new questions. For instance, by showing the model that "Who was the first person to walk on the moon?" corresponds to an "ordinal constraint," the model can learn to identify similar ordinal logic in other questions.
> - **Semantic Parsing to Logical Forms** This approach involves translating the natural language question into an intermediate, machine-readable logical form (e.g., λ-Calculus or a SPARQL query sketch). The resulting logical form inherently and explicitly contains the core constraints of the question. For example, the question "Which films were directed by Ang Lee and starred Chow Yun Fat?" could be parsed into `λx.(directed_by(x, Ang_Lee) ∧ starred_by(x, Chow_Yun_Fat))`. By analyzing this logical form, we can directly and accurately extract the two "multi-entity constraints."
> - **KG** **Schema and** **Ontology** **Analysis** This method directly utilizes the structured *a priori* knowledge defined within the KG itself. By analyzing the KG's schema or ontology, we can automatically extract the rules embedded within. For instance, if a relation `award_year` has a "domain" of "Award" and a "range" of "Year," then any question involving this relation can be automatically associated with a type constraint specifying that "the answer must be a year." This is a high-precision method, though its coverage is limited by the completeness of the KG schema.
>
> In more complex domains such as law, finance, or medicine, a question's constraints are often more nuanced and implicit. The methods above can work in synergy to address these challenges:
>
> - **Handling Implicit Constraints**: For questions like, "What public offices did Roosevelt hold before he became president?", the temporal constraint ("before he became president") is relative and implicit. **LLM-based induction** is particularly effective in such scenarios due to its superior ability to understand the contextual and temporal logic of natural language.
> - **Ensuring Logical Rigor**: In domains requiring high precision, such as law or finance, **semantic parsing** can translate complex clauses or questions into unambiguous logical forms. This ensures all conditions are captured accurately, avoiding reasoning errors that can arise from the ambiguity of natural language.
> - **Leveraging Domain-Specific Priors**: In specialized KGs (e.g., a biomedical KG), the schema is often highly detailed. **KG schema analysis** can extract a large number of high-confidence, domain-specific constraints (e.g., "Protein A *participates in* Biological Process B"), providing a solid foundation of domain knowledge for the reasoning process.
>
> **A2** Based on our experiments and case analyses (as illustrated in Figures 9 and 10), the failures of the DP framework during the path generation phase can be attributed to the following key factors:
>
> - **Subtle Structural Variations in Complex Reasoning Scenarios**: In some challenging reasoning cases, incorrect paths may exhibit high structural similarity to the ground-truth paths, differing only in a critical relation. For example, in the failure case shown in Figure 9, the model-generated path is nearly identical to the correct one, but a minor deviation ultimately leads to reasoning failure. This suggests that although our progressive knowledge distillation strategy (including SFT and KTO) effectively equips the LLM with a strong understanding of structural patterns in the knowledge graph, it still struggles to distinguish between such "high-fidelity" incorrect paths. A potential reason is the lack of sufficiently diverse and contrastive negative samples, which limits the model's ability to capture decisive semantic differences.
> - **Error Accumulation in Multi-Hop Reasoning**: For tasks requiring long-chain reasoning, minor deviations in early steps of path generation can compound and result in significant divergence from the correct path. Although the DP framework incorporates introspection and backtracking mechanisms for error correction, their effectiveness is constrained when the initially generated candidate path set is of low overall quality. In such cases, even the subsequent selection and verification processes may fail to identify a completely correct path.
>
> **Future Directions**:
>
> - **Developing Minimal Yet Informative Negative Sampling:** Rather than introducing overly complex or synthetic negative examples, we aim to design *simple but impactful* negative sampling strategies. For instance, we can mine naturally occurring “borderline” paths within the KG that deviate from gold paths by only one relation or entity. These naturally occurring “near-miss” paths not only reflect realistic ambiguity but also improve the model’s robustness in a lightweight manner. This approach balances discriminative training and implementation simplicity.
> - **Enabling Lightweight and Targeted Path Revision:** Instead of overhauling the entire reasoning path upon constraint violation, we advocate for a **modular, token-level correction mechanism**. This involves allowing the model to revise *only* the faulty component—such as a misaligned relation or misplaced intermediate entity—while retaining valid portions of the path. This incremental correction strategy reduces computational overhead, encourages reusability of partial reasoning, and fits well into few-shot or constrained-interaction settings.
>
> **A3**  DP is intentionally designed with several mechanisms that enable it to handle the aforementioned issues to a certain extent:
>
> - **Decoupling** **of Path Planning and Instantiation**: The DP framework first employs an LLM to generate candidate relation paths and then proceeds to instantiate these paths within the KG to identify concrete entities and facts. This decoupling provides several inherent advantages:
>   - **Handling Incomplete Knowledge**: If a particular fact is missing from the KG, the corresponding path will fail during instantiation and be discarded during the verification phase. The framework then attempts to instantiate alternative candidate paths. Since the path generator typically outputs multiple candidates, this increases the likelihood of discovering a viable reasoning route even under incomplete knowledge conditions.
>   - **Mitigating Noisy Knowledge**: If a spurious fact exists in the KG, leading to an incorrect yet superficially plausible reasoning path, the introspection module plays a crucial role. It evaluates each path against constraints extracted from the question (e.g., type constraints, multi-entity constraints). If the noisy path violates these constraints, it is identified and filtered out, triggering the framework’s backtracking mechanism.
> - **Multiple Candidate Paths with Semantic Selection**: The framework generates multiple candidate paths and applies semantic relevance filtering to prioritize plausible ones. This design ensures that even if some paths are invalidated due to KG noise or incompleteness, the system can still fall back on other semantically valid and executable alternatives within the KG, increasing the robustness of answer retrieval.
>
> **Remaining Challenges and Future Directions**:
>
> Despite these design advantages, we acknowledge that the DP framework faces limitations when deployed on large-scale, noisy, and highly incomplete real-world KGs:
>
> - **Collective Failure of Candidate Paths**: In scenarios where critical portions of the KG are severely missing, all generated candidate paths may fail during instantiation, resulting in the inability to derive an answer.
> - **Constraint-Evasive Noise**: In rare edge cases, an erroneous fact may structurally satisfy all predefined constraints, allowing it to pass through the introspection mechanism undetected.
>
> While such extreme conditions are uncommon, they highlight important areas for further enhancement:
>
> - **Integrating** **Knowledge Graph** **Completion**: Although KG completion is a separate research domain, it can be incorporated into our pipeline. During the path instantiation stage, if a path fails due to missing links, a KG completion model can be invoked to predict the missing relations and potentially “repair” the broken reasoning chain.
> - **Incorporating Fact Verification Mechanisms**: Before finalizing an answer, a fact verification step can be introduced to cross-check key facts within the reasoning path using external textual corpora or more trustworthy sources, thereby improving the framework’s resilience to noise.
>
> In summary, DP demonstrates a degree of robustness in handling imperfect knowledge graphs through its modular design. However, continued research is necessary to further enhance its performance in complex, real-world applications.

---

> > ### Comment · Reviewer_iFUY · 2025-08-04
> >
> > Thank you for your detailed reply. Please consider strengthening the paper’s organization, and experimental comparisons in the future version. I will keep my current score in this round.

---

> > > ### Author Response · Authors · 2025-08-04
> > >
> > > Thanks for your recognition of this paper. We appreciate your suggestion regarding the paper's organization and experimental comparisons, and we will carefully revise these aspects in the future version to further improve the clarity and strength of our work. We are grateful for your time and thoughtful review.

---

### Note · Authors · 2025-08-14

Dear Area Chairs and Reviewers,

We sincerely thank you for your thoughtful reviews and constructive feedback on our paper. Your insightful comments have helped us significantly improve the clarity, rigor, and presentation of our work.

In response to your suggestions during the rebuttal phase, we have made the following key enhancements:

1. We have conducted statistical significance tests to rigorously evaluate the performance differences between our method (DP) and the baselines.

2. We have expanded the case study with additional analysis and interpretation to provide deeper qualitative insights.

3. To ensure fair and comprehensive comparisons, we have added experiments under consistent evaluation settings such as on the full test sets across all datasets.

4. To better highlight the effectiveness of DP across diverse models, we report results for all evaluated LLMs on all three datasets.

We are grateful that all reviewers confirmed that their concerns have been adequately addressed, and several have indicated their intention to update their scores accordingly.

In the camera-ready version, we will carefully incorporate all remaining suggestions to further refine the paper’s presentation and readability.

Thank you once again for your time and valuable feedback.

Best regards,

The Authors

---

### Decision · Program_Chairs · 2025-09-17

**Decision:**

Accept (poster)

**Comment:**

(a) Summarize the scientific claims and findings of the paper based on your own reading and characterizations from the reviewers.
This paper introduces "Deliberation on Priors" (DP), a novel framework designed to enhance the trustworthiness of Large Language Models (LLMs) when reasoning over Knowledge Graphs (KGs). The primary claim is that by explicitly leveraging both structural and logical "priors" from the KG, the framework can mitigate LLM hallucinations and improve the faithfulness and reliability of generated responses. The DP framework operates in two main stages: (1) a Progressive Knowledge Distillation stage that integrates the KG's structural priors into the LLM via supervised fine-tuning (SFT) and Kahneman-Tversky Optimization (KTO), teaching the model to generate structurally valid reasoning paths; and (2) a Reasoning-Introspection stage where the LLM uses extracted logical constraint priors from the question to verify, filter, and refine these paths via a backtracking mechanism. The authors demonstrate through extensive experiments on three benchmark KGQA datasets (WebQSP, CWQ, MetaQA) that DP achieves new state-of-the-art results, significantly outperforming existing methods in accuracy and faithfulness, while also being more efficient in terms of LLM API calls.

(b) What are the strengths of the paper?
The paper has several strengths, as also highlighted by the reviewers:

High Relevance and Novelty: The work addresses the critical and timely problem of LLM hallucination in the context of retrieval-augmented generation. The proposed DP framework is a novel and well-motivated contribution, cleverly combining knowledge distillation of structural information with a symbolic, constraint-based verification loop.

Strong Empirical Results: The method demonstrates substantial performance gains over strong baselines across multiple datasets and LLMs. The authors' claims of state-of-the-art performance are well-supported by the evidence, which was further strengthened during the rebuttal period with statistical significance tests.

Methodological Rigor and Clarity: The paper is well-written, clearly structured, and easy to follow. The proposed framework is explained in detail, and the ablation studies are comprehensive, effectively isolating and validating the contribution of each component of the DP framework.

Practical Considerations: The authors show awareness of real-world deployment constraints by analyzing and reporting on the efficiency of their method, noting that DP requires significantly fewer LLM calls than competing iterative reasoning approaches.

(c) What are the weaknesses of the paper? What might be missing in the submission?
The initial submission had a few weaknesses, which were largely addressed during the rebuttal but are still worth noting as limitations of the approach:

Reliance on Predefined Constraints: The framework's introspection module, in its highest-performing configuration, relies on a set of manually defined constraint types. This limits its out-of-the-box generalizability to new domains where such constraints are not readily available. While the authors demonstrated a feasible automated alternative, it came with a performance trade-off.

KG-Specific Fine-tuning: The approach requires a fine-tuning stage for each target KG. While the authors argue this is efficient due to LoRA, it still presents a scalability hurdle compared to purely zero-shot or in-context learning methods, making it less suitable for applications that need to query many different KGs dynamically.

Dependence on KG Quality: Like all KG-based reasoning methods, DP's performance is inherently capped by the quality, completeness, and accuracy of the underlying knowledge graph.

(d) Provide the most important reasons for your decision to accept/reject. For spotlights or orals explain why the paper stands out (other than by high scores or popularity trends).
The primary reason for acceptance is that this is a technically solid paper that makes a valuable contribution to an important area of research. The strengths—a novel and effective framework, strong and now rigorously validated empirical results, and clear presentation—outweigh the acknowledged limitations.

The authors engaged with the feedback, conducting extensive new experiments to address reviewer concerns about evaluation fairness, statistical significance, and baseline comparisons. This strengthened the paper, its scientific rigor, and the validity of the work. The final paper, incorporating the results of these discussions, will be a valuable addition to the NeurIPS program.

(e) Summarize the discussion and changes during the rebuttal period. What were the points raised by the reviewers? How were each of these points addressed by the authors? How did you weigh in each point in your final decision? (Do not mention reviewer names. Use their anon ids instead.)
The discussion period was productive and led to significant improvements in the paper's evaluation and claims. The main points raised and addressed were:

Manual Constraints (raised by iFUY, K6r7, BKJv): Reviewers were concerned about the scalability of relying on manually defined constraints. The authors addressed this by pointing to an ablation study in the paper that used an automated, prompt-based approach and showed it was still effective, though slightly less so than the manual version. They also detailed several promising avenues for future work on automatic constraint extraction, which satisfied the reviewers (e.g., K6r7).

Evaluation Fairness and Robustness (raised by jD5U, K6r7, BKJv): This was the most critical area of discussion.

Sampled Test Sets: Reviewer BKJv questioned the use of sampled test sets. The authors responded by first re-evaluating a key baseline on their exact sampled set to ensure a fair head-to-head comparison. Following further discussion, they went above and beyond by re-running their method on the full test sets for WebQSP and CWQ, demonstrating that their method remains state-of-the-art. This was a crucial change that significantly strengthened the paper's claims.

Inconsistent Base LLMs: Reviewer BKJv pointed out that comparing methods using different underlying LLMs was problematic. The authors provided new results comparing DP with other supervised methods using the exact same LLaMA2-Chat-7B model, as well as a comprehensive table comparing naive LLM performance vs. DP-enhanced performance for a wide range of models.

Statistical Significance: Reviewer jD5U correctly noted the absence of statistical tests for the results. In response, the authors ran their model 10 times, performed one-tailed t-tests against the best baselines, and confirmed that their improvements were highly statistically significant (p<0.001). They committed to updating the tables to only bold results that meet this criterion.

Scalability due to Fine-Tuning (raised by BKJv): The authors defended the necessity of fine-tuning for achieving high faithfulness and argued that their use of LoRA makes it computationally efficient. They also provided specific details on the computational cost.

The authors' thorough responses grounded in empirical results led reviewers jD5U, K6r7, and BKJv to explicitly state their concerns were addressed and raise their scores. This engagement and the resulting improvements were a major factor in my final recommendation to accept the paper. The paper is now substantially stronger and more convincing as a result of the review process.